# Molecular basis for the enzymatic inactivity of class III glutaredoxin ROXY9 on standard glutathionylated substrates

Pascal Mrozek [1], Stephan Grunewald [1], Katrin Treffon[1], Gereon Poschmann [2], Fabian Rabe von Pappenheim [3,4], Kai Tittmann [3,4] & Christiane Gatz [1] ✉

Class I glutaredoxins (GRXs) are nearly ubiquitous proteins that catalyse the glutathione (GSH)-dependent reduction of mainly glutathionylated substrates. In land plants, a third class of GRXs has evolved (class III). Class III GRXs regulate the activity of TGA transcription factors through yet unexplored mechanisms. Here we show that *Arabidopsis thaliana* class III GRX ROXY9 is inactive as an oxidoreductase on widely used model substrates. Glutathionylation of the active site cysteine, a prerequisite for enzymatic activity, occurs only under highly oxidizing conditions established by the GSH/glutathione disulfide (GSSG) redox couple, while class I GRXs are readily glutathionylated even at very negative GSH/GSSG redox potentials. Thus, structural alterations in the GSH binding site leading to an altered GSH binding mode likely explain the enzymatic inactivity of ROXY9. This might have evolved to avoid overlapping functions with class I GRXs and raises questions of whether ROXY9 regulates TGA substrates through redox regulation.

Glutaredoxins (GRXs) have been described as either thiol-disulfide oxidoreductases (class I) or iron-sulfur (FeS) cluster binding proteins (class II)[2]. They are not only involved in fundamental cellular functions like DNA synthesis and FeS cluster transfer in both pro- and eukaryotes, but also in the regulation of different key proteins that coordinate cellular processes (for reviews see e.g[3,4].). Under conditions of oxidative stress, class I GRXs are assumed to protect reactive thiol groups of proteins from irreversible oxidation by catalysing their glutathionylation and subsequent de-glutathionylation[5,6].

As summarized in several reviews[7–11], GRXs are characterized by a thioredoxin fold which consists of a central four-stranded β-sheet surrounded by three α-helices. They share a conserved 'active site' at the beginning of helix 1 of the thioredoxin fold. The 'active site' is a variant of the sequence CPYC in class I GRXs and a very conserved CGFS motif in class II GRXs. GRXs interact with the tripeptide glutathione (GSH), which serves as an electron donor for the reduction of disulfides by class I GRXs or as a co-factor to coordinate FeS clusters in class II GRXs. When functioning as thiol-disulfide oxidoreductases, GRXs can operate like thioredoxins in reducing disulfide bridges by forming a mixed disulfide between the catalytic cysteine of the active site ($Cys_A$) and the client protein. This can either be resolved by the second cysteine ($Cys_B$) in the active center (dithiol mechanism) or by GSH (monothiol mechanism)[12]. The disulfide within the active site is subsequently reduced through a glutathionylated intermediate by in total two molecules GSH leading to the release of glutathione disulfide (GSSG). When functioning as a reductase of glutathionylated substrates, the glutathione moiety of the substrate has to be positioned into the GSH binding groove so that the sulphur atom points directly towards the thiol group of $Cys_A$[13,14]. The specific orientation within this so-called scaffold binding site allows the transfer of glutathione from

[1]Department of Plant Molecular Biology and Physiology, Albrecht-von-Haller Institute for Plant Sciences, Georg-August-University Göttingen, Julia-Lermontowa-Weg 3, 37077 Göttingen, Germany. [2]Institute of Molecular Medicine, Proteome Research, Medical Faculty and University Hospital, Heinrich-Heine-University Düsseldorf, Universitätsstraße 1, 40225 Düsseldorf, Germany. [3]Department of Molecular Enzymology, Göttingen Centre for Molecular Biosciences and Albrecht-von-Haller-Institute, Georg-August-University Göttingen, Julia-Lermontowa-Weg 3, 37077 Göttingen, Germany. [4]Max-Planck-Institute for Multidisciplinary Sciences, Am Faßberg 11, 37077 Göttingen, Germany. ✉e-mail: cgatz@gwdg.de

glutathionylated substrates to $Cys_A$, resulting in glutathionylated GRXs and the release of the reduced substrate. Glutathionylated GRXs are subsequently reduced by a second molecule of GSH, which is recruited by the so-called activator site[13].

In contrast to class I GRXs, class II GRXs have no reductase activity in enzyme assays with low molecular weight substrates bis(2-hydroxyethyl)disulfide (HED) or L-cysteine-glutathione-disulfide (Cys-SG)[13]. Structure-function analysis of class I and class II GRXs have shown that a conserved five amino acid long loop located between a highly conserved lysine residue and $Cys_A$ is the major determinant that interferes with oxidoreductase activity[14,15]. This loop shifts the GSH thiol group away from $Cys_A$ allowing the thiol groups of GSH and $Cys_A$ to coordinate a labile FeS cluster in a cluster-bridged dimeric holoprotein. Class I GRXs with the active site variants CSYC or CGYC rather than CPYC[16] and also some CPYC-encoding GRXs can also bind FeS clusters[17–20]. The FeS-containing class I holoproteins are characterized by an increased stability and different mode of dimerization as compared to the holoproteins from class II GRXs[14].

Land plants yet contain a third class of GRXs (class III or CC-type GRXs)[21]. The gene family of class III GRXs has expanded during land plant evolution and contains 21 members (ROXY1-21) in the model plant *Arabidopsis thaliana*[22]. According to protein structure predictions[23], they also adopt the thioredoxin fold, which puts the putative active site, a CCMC/S or CCLC/S motif, at the beginning of helix 1 (shown exemplarily for ROXY9 in Fig. 1a). Previous structural studies of class I and class II GRXs from different organisms had identified several amino acid residues that are involved in glutathione binding[13,14]. The amino acid environments of these residues as found in sequences representing all three GRX classes encoded in the Arabidopsis genome are shown in Fig. 1b. The alignment highlights that class III GRXs do not encode the class II-specific five amino acid loop which interferes with oxidoreductase activity[14,15], nor the proline in the active site which might interfere with FeS cluster assembly[16].

Due to the redundancy of closely related members of this large gene family, only few robust loss-of-function phenotypes are known. A role in flower development was shown for class III GRXs ROXY1 and ROXY2[24,25], while ROXY6, ROXY8 and ROXY9 (also called CEPD1, CEPD1-like1 and CEPD2) are mobile shoot to root signals which are necessary for activation of nitrate uptake genes upon nitrogen starvation[26]. In general, the picture has emerged that the unique function of class III GRXs is to regulate the activity of TGACG-binding (TGA) transcription factors[25,27–29]. The maize class III GRX MSCA1 regulates meristem size presumably by controlling the redox state and thus the DNA binding activity of TGA transcription factor FEA4[28]. Still, the molecular mechanisms underlying the regulation of other class III GRXs/TGA couples have remained elusive.

While a vast amount of biochemical and structural analyses was performed for class I and class II GRXs from model organisms like e.g. *Escherichia coli*, *Saccharomyces cerevisiae*, *Plasmodium falciparum*, *Chlamydomonas reinhardtii*, poplar, zebrafish, mice, humans etc., almost no information is available for class III GRXs. This has been due to encountered difficulties when purifying recombinant proteins expressed in *E. coli*[30]. Here, we succeeded in obtaining milligram amounts of class III GRX ROXY9 from *Arabidopsis thaliana* by applying the baculovirus expression system in insect cells.

In this work, we show that ROXY9 has weak GSH-dependent deglutathionylation activity on glyceraldehyde-3-phosphate dehydrogenase (GAPDH), but no oxidase or reductase activities on other substrates like HED, Cys-SG, cumene hydroperoxide (CHP) or roGFP2. In contrast to class I GRXs, which are glutathionylated at $Cys_A$ over a wide range of GSH/GSSG redox potentials, only the internal disulfide-containing oxidized form is detected for ROXY9 at a midpoint redox potential between −220 and −230 mV at p$H$ 7.0. Since GSH-dependent redox reactions require the glutathionylated intermediate, we explain the lack of efficient oxidoreductase activity on glutathionylated substrates by a different GSH binding mode that possibly inflicts strain on the disulfide between ROXY9 and glutathione.

## Results

### ROXY9 adopts a thioredoxin fold similar to Arabidopsis GRXC2

Class III GRX ROXY9 (AT2G47880) and class I GRXC2 (AT5G40370) from *A. thaliana*, the latter serving as a positive control in all our assays, were expressed in insect cells as fusions with a strep-tagged maltose binding protein (strep-MBP). After affinity purification in the presence of 5 mM GSH (Supplementary Fig. 1a), 25 to 40 mg of both proteins were routinely obtained from a 600 ml culture of Hi5 cells. Despite the high protein concentration (100 to 200 μM), samples lacked the brownish colour indicative for FeS clusters. Analytical gel filtration analysis documented that at least half of the proteins eluted as aggregates/higher oligomers of > 670 kDa (Supplementary Fig. 1b-d). The non-aggregated protein species appeared to elute as monomers, dimers and trimers in case of strep-MBP-ROXY9 (Supplementary Fig. 1b) and as monomers in case of strep-MBP-GRXC2 (Supplementary Fig. 1c). DTT treatment of strep-MBP-ROXY9 led to a shift from oligomers to mostly monomers (Supplemental Fig. 1d), indicating that the protein is prone to forming intermolecular disulfides even in the presence of 5 mM GSH. The very high molecular weight aggregates could not be resolved by treatment with DTT.

To investigate whether ROXY9 and GRXC2 adopt a similar protein fold, ROXY9 and GRXC2 were first analysed by far-UV CD spectroscopy. To this aim, the strep-MBP tag was removed by cleavage with Tobacco Etch Virus (TEV) protease and untagged ROXY9 and GRXC2 were purified by gel filtration (Supplementary Fig. 1e,f). The CD spectra of both proteins were almost identical, independent of whether measurements were made in the presence of DTT or GSH (Fig. 1c). Both proteins denatured at a melting temperature of 70.6°C and 69.2°C, respectively (Supplementary Fig. 2a), which is consistent with the previously reported high thermal stability of other GRXs[14,31]. Whereas GRXC2 refolded when the temperature was lowered, thermal unfolding of ROXY9 was irreversible (Supplementary Fig. 2b). Overall, we conclude from these experiments that ROXY9 most likely adopts a canonical thioredoxin fold akin to GRXC2.

The predicted thioredoxin fold of ROXY9 positions the putative redox active cysteines of the $C_{21}CLC_{24}$ motif in a way that an intramolecular disulfide can be formed between Cys21 and Cys24, similar to the disulfide identified in CPYC-type GRXs[32,33] (Fig. 1a). Typically, the catalytic cysteine is exposed to the solvent, while the resolving cysteine is buried, a pattern that is also observed for GRXC2 and ROXY9 (Supplementary Table 1). To provide experimental evidence for the existence of this disulfide and to determine its midpoint redox potential at p$H$ 7.0, strep-MBP-ROXY9 was incubated with different ratios of DTT/dithiane, which—as calculated by the Nernst equation—translates into redox potentials between −290 and −210 mV at this p$H$. The redox states were monitored and quantified by alkylation of free thiol groups with 5 kDa methoxy maleimide polyethylene glycol (mmPEG) and subsequent analysis of the protein by non-reducing SDS polyacrylamide gel electrophoresis (PAGE)[33,34]. Upon treatment of strep-MBP-ROXY9 with 10 mM DTT and subsequent alkylation of the TCA-precipitated protein in the presence of 1% SDS, the mobility of the protein was reduced due to the addition of mmPEG to the five reduced cysteines in the ROXY9 moiety of the protein (Fig. 2). The shift was larger than expected, a phenomenon that has been described before and might be due to the interaction of mmPEG with the polyacrylamide matrix[33]. Under more oxidative conditions, a second band with higher mobility appeared. Moreover, the amount of protein species with very low electrophoretic mobility increased, again demonstrating the tendency of the protein to form intermolecular disulfides as already revealed by size exclusion chromatography (Supplementary Fig. 1). The reduced and the oxidized species of strep-MBP-ROXY9 were present in roughly the same amounts at a redox

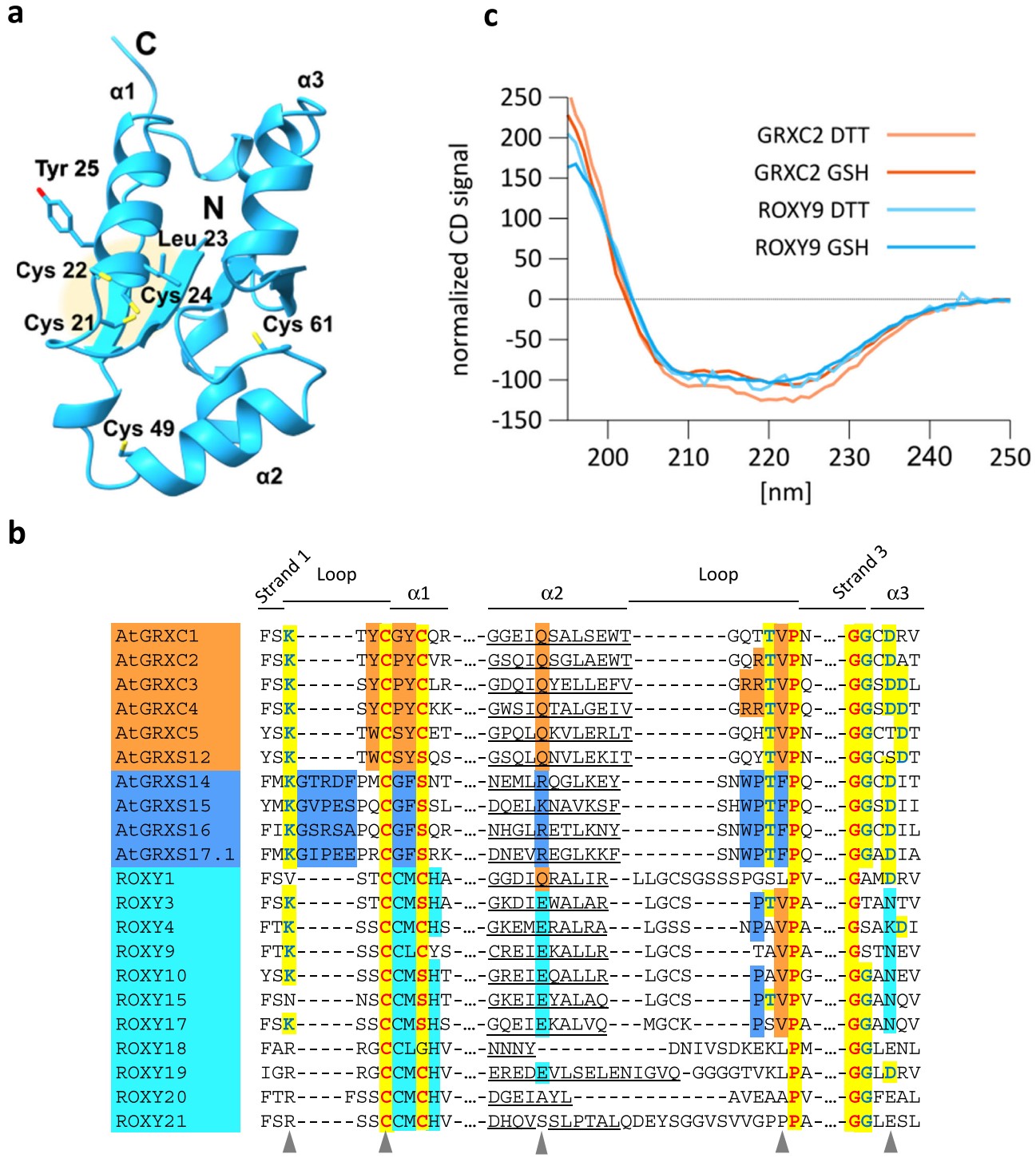

**Fig. 1 | ROXY9 and GRXC2 share the thioredoxin fold. a** Model of ROXY9 according to AlphaFold. Side chains of the five cysteines, the leucine within and the tyrosine adjacent to the CCLC motif are shown. **b** Alignment of Arabidopsis GRX sequences facing the GSH binding grove. Colours indicate different degrees of sequence conservation. Red letters on yellow background: highly conserved in all three classes of GRXs; Blue letters on yellow background: conserved in class I and class II GRXs; dark orange background: conserved only in class I GRXs; blue background: conserved in class II GRXs, cyan background: conserved in class III GRXs. Amino acids forming α helix 2 according to AlphaFold are underlined. Triangles indicate amino acids that have been shown to contact GSH in at least one solved structure. The figure was adapted from Begas et al., 2017 and Liedgens et al., 2020. **c** Far-UV CD spectra of unfused ROXY9 and GRXC2 in the 190-250 nm range. Proteins were either in 1 mM DTT or in 0.5 mM GSH.

potential between −230 and −240 mV at pH 7. This is in the range of the midpoint redox potentials of intramolecular disulfide bridges within the active sites of class I GRXs, which vary between −198 and −263 mV at this pH[33,35,36]. For the corresponding disulfide of strep-MBP-GRXC2, the midpoint redox potential was also found to range between −230

and −240 mV. Incubation with GSSG led to further oxidation of both proteins presumably due to glutathionylation or other oxidations of cysteines outside the active site.

In order to provide further evidence that the oxidized species of strep-MBP-ROXY9 indeed contains the expected disulfide between

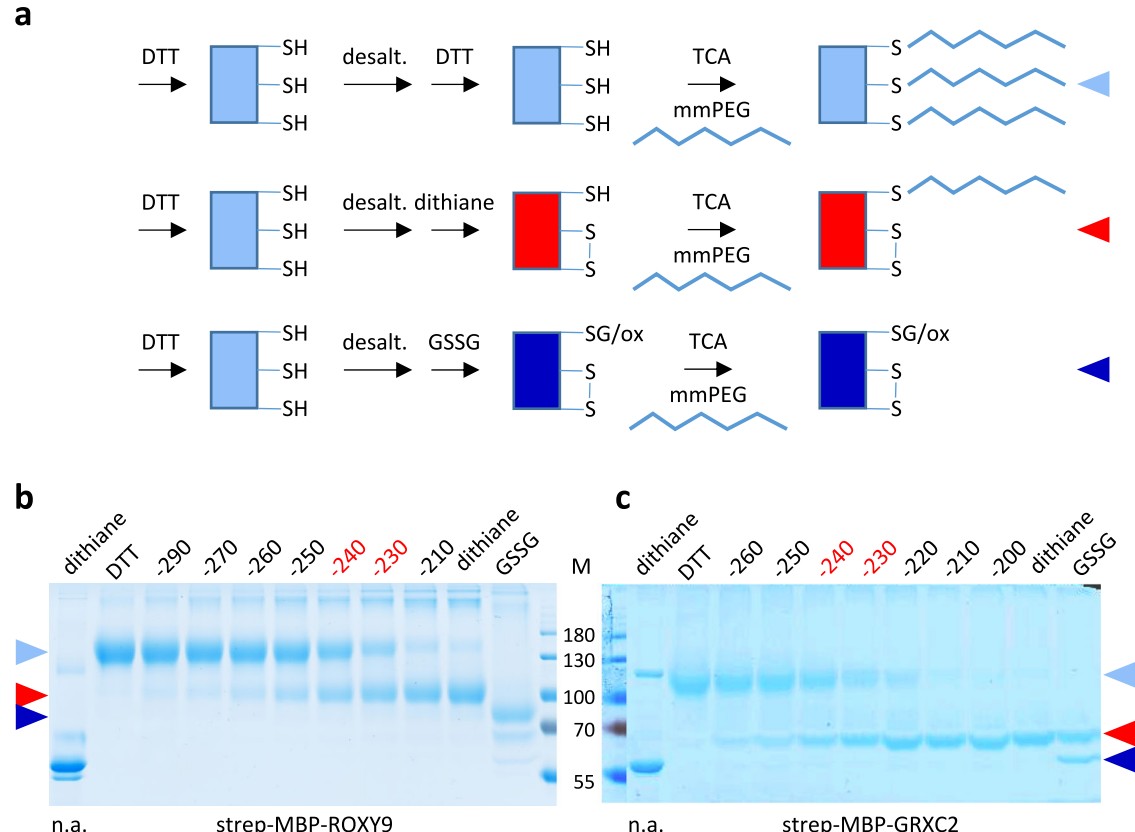

**Fig. 2 | The redox potentials of strep-MBP-ROXY9 and strep-MBP-GRXC2 are similar. a** Sketch of the alkylation experiment. Proteins (approximately 0.2 mM) were pre-reduced with 10 mM DTT, desalted and mixed with different DTT/dithiane (10 mM) redox buffers establishing the indicated redox potentials (mV) or with 10 mM GSSG. After TCA precipitation, reduced cysteines were labelled with 5 kDa mmPEG. The expected alkylation pattern of a reduced and oxidized GRX with an active site CxxC motif and an additional cysteine is shown. **b**, **c** Non-reducing SDS PAGE of alkylated strep-MBP-ROXY9 and strep-MBP-GRXC2. Midpoint redox potentials are indicated in red. The first lane (n.a.) contains the respective protein after dithiane treatment, TCA precipitation and incubation in alkylation buffer lacking mmPEG. For documentation of the mobility of non-alkylated strep-MBP-ROXY9 on this gel after different treatments, see Supplementary Fig. 3. The triangles denote the different redox states: light blue: fully reduced; red: disulfide containing; dark blue: fully oxidized. Molecular masses of marker proteins (M) are indicated in kDa. Uncropped gels are provided as a Source Data file.

Cys21 and Cys24, strep-MBP-ROXY9 variants with either SCLC or CCLS sequences in the 'active site' were analysed (Supplementary Fig. 4a,b). The majority of these protein variants remained in the reduced form even in the presence of dithiane only. This is consistent with the notion that the band with higher mobility reflects strep-MBP-ROXY9 harbouring an intramolecular disulfide between Cys21 and Cys24 (ROXY9-$S_2$), which cannot be formed by these variants. Analysis of the strep-MBP-ROXY9-CSLC variant showed that the class III-specific second cysteine does not influence the midpoint redox potential of the Cys21-Cys24 disulfide at p$H$ 7 (Supplementary Fig. 4c). We conclude from these experiments, that strep-MBP-ROXY9 expressed in insect cells adopts a thioredoxin fold allowing formation of a disulfide between Cys21 and Cys24 within the $C_{21}$CLC$_{24}$ signature.

**Strep-MBP-ROXY9 has no oxidoreductase activity with HED, Cys-SG or roGFP2 but can weakly de-glutathionylate GAPDH-SG in a GSH-dependent manner**

Enzymatically active GRXs use glutathionylated molecules as substrates[37,38]. One of the standard assays for testing oxidoreductase activity of GRXs is the HED (or HEDS) assay[39–42]. In this assay, GRXs first catalyse the reduction of the disulfide in HED with GSH as electron donor, leading to ß-mercaptoethanol (ß-ME) and glutathionylated ß-ME (ß-ME-SG), the latter serving as a substrate for the de-glutathionylation reaction[42]. GSSG is subsequently reduced by glutathione reductase (GR) in the presence of NADPH, the consumption of which is measured spectrophotometrically. After adding strep-MBP-

GRXC2 to the reaction mixture, absorption at 340 nm decreased, indirectly indicating that GSSG was generated during GRXC2-catalyzed reduction of HED (Fig. 3a). In contrast, strep-MBP-ROXY9 and also untagged ROXY9 did not exhibit any detectable enzymatic activity (Fig. 3a, Supplementary Fig. 5). Since several class I GRXs become more active when mutated for their resolving cysteine residues[43–45], we changed the ROXY9 CCLC signature into CCLS. We also generated a strep-MBP-ROXY9-CPYC derivative in order to ask the question whether the class III GRX-specific second cysteine might interfere with catalytic activity. However, these alterations did not rescue enzymatic activity (Fig. 3a). Likewise, Strep-MBP-ROXY9 and its derivatives were unable to de-glutathionylate Cys-SG (Fig. 3b).

Since it was previously reported that cumene hydroperoxide (CHP) can be reduced by members of the poplar class III GRX gene family[46], we tested strep-MBP-ROXY9 and strep-MBP-GRXC2 for cumene hydroperoxidase activity (Supplementary Fig. 6). The assay was carried out similar to the HED assay with HED being replaced as substrate by CHP. In our hands, consumption of NADPH was independent of the presence of strep-MBP-ROXY9 or strep-MBP-ROXY9-CPYC. Only a slightly steeper descent was measured in the presence of strep-MBP-GRXC2. We thus conclude that CHP is not a substrate for strep-MBP-ROXY9 and only a poor substrate for strep-MBP-GRXC2.

With ROXY9 showing no measurable reductase activity towards low molecular weight substrates HED, Cys-SG or CHP, our focus shifted to roGFP2 as a potential protein substrate. Oxidized roGFP2 contains a disulfide (roGFP2-$S_2$), which is reduced by GSH resulting in

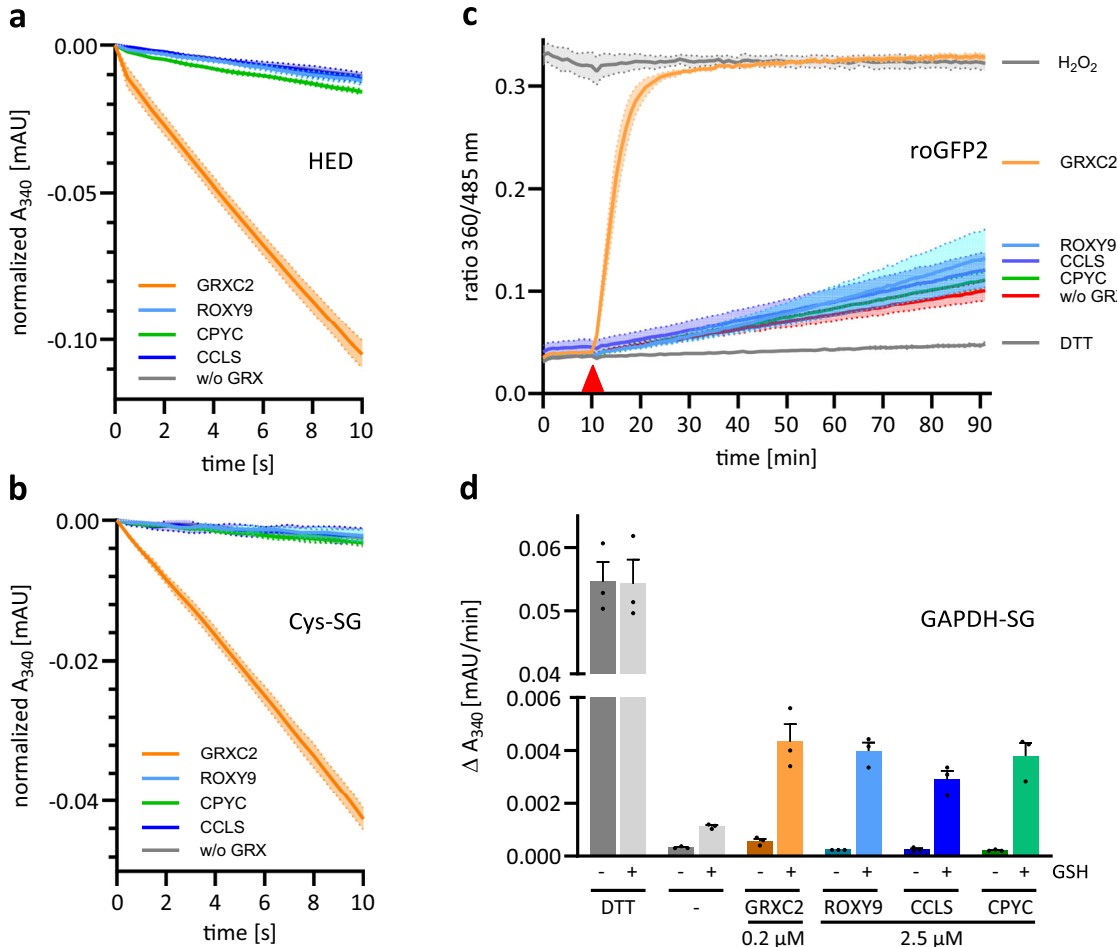

**Fig. 3 | ROXY9 has no oxidoreductase activity on standard substrates but weak GSH-dependent reductase activity on glutathionylated GAPDH. a** HED and **b** Cys-SG assays with strep-MBP-GRXC2, Strep-MBP-ROXY9, strep-MBP-ROXY9-CCLS and strep-MBP-ROXY9-CPYC (1.6 μM each). **c** roGFP2 assay with strep-MBP-GRXC2, strep-MBP-ROXY9, strep-MBP-ROXY9-CCLS and strep-MBP-ROXY9-CPYC. Final concentrations were 2 μM roGFP2, 2 μM GRX and 50 μM GSSG, which was added at 10 min (red triangle). References were treated with DTT or H$_2$O$_2$, respectively, to represent fully reduced or oxidized roGFP2. See Supplementary Fig. 8 for data on the influence of GRXs on the roGFP2 redox state in the absence of GSSG. **d** Reactivation of GAPDH activity. Enzymatically inactive glutathionylated

GAPDH (GAPDH-SG) (0.6 μM) was incubated with either DTT, strep-MBP-GRXC2 (0.2 μM), strep-MBP-ROXY9, strep-MBP-ROXY9-CCLS or strep-MBP-ROXY9-CPYC (each 2.5 μM) in the absence or presence of 2 mM GSH for 20 min. Subsequently, consumption of NADH per min upon reduction of 1,3 bisphosphoglycerate catalysed by GAPDH was monitored at 340 nm. Error bars or error bands represent the standard error of the mean resulting from three (**a, c, d**) or two (**b**) biological replicates, each replicate being represented by an independent protein preparation. The mean of three (**c**) or four (**a, b, d**) measurements (technical replicates) was calculated for each biological replicate. Source data are provided as a Source Data file.

glutathionylated roGFP2 (roGFP2-SG)[14,47]. Reduction of roGFP2-SG by GSH is the rate limiting step which is accelerated by GRXs, with its reactive cysteine attacking the disulfide of roGFP2-SG, resulting in glutathionylated GRX (GRX-SG) and roGFP2. GRX-SG is then reduced in the presence of GSH, GR and NADPH. The redox state of roGFP2 is determined by measuring its fluorescence emission at 520 nm after excitation at two wavelengths that distinguish between the reduced and the oxidized form (360 nm and 485 nm)[48]. As described before for His-tagged GRXC2[49], strep-MBP-GRXC2 reduced pre-oxidized roGFP2 in the presence of GSH, NADPH and GR, while strep-MBP-ROXY9 failed to catalyse this reaction (Supplementary Fig. 7).

The roGFP2 assay offers the advantage, that the putativie oxidase activity of GRXs can also be tested. Even class II GRXs AtGRXS15, AtGRXS16 and HsGRX5 can catalyse the oxidation of roGFP2 by GSSG, although they fail to mediate its reduction[13,14,50]. GRX-mediated oxidation by GSSG starts with the glutathionylation of the reactive cysteine of GRX (GRX-SG). The reactive thiol of roGFP2 attacks this disulfide and glutathione is transferred to roGFP2 by a disulfide exchange reaction. The resolving cysteine of roGFP2 attacks the disulfide of roGFP2-SG leading to formation of an intramolecular

disulfide[14,47]. As expected, oxidation of roGFP2 by GSSG was catalysed by strep-MBP-GRXC2 (Fig. 3c). In contrast, strep-MBP-ROXY9 and its CPYC and CCLS variants were inactive. Since GRXs and roGFP2 are present in equimolar amounts in the assay, we tested, whether roGFP2 forms an unresolved mixed disulfide with strep-MBP-ROXY9. However, analysis of the reaction mixture on a non-reducing SDS PAGE clearly excluded this scenario (Supplementary Fig. 9).

Considering, that even class II GRX3 from *Chlamodymonas reinhardtii*, which fails to reduce HED, is able to reduce glutathionylated glyceraldehyde-3-phosphate dehydrogenase (GAPDH-SG)[41], we asked whether ROXY9 might show enzymatic activity in this assay. In order to measure de-glutathionylation of GAPDH, the enzyme is first glutathionylated and thus inactivated in its enzymatic function. Subsequently, GAPDH-SG is incubated in the presence of GRXs and GSH. Finally, GAPDH-catalyzed reduction of its substrate 1,3 bisphosphoglycerate with the co-substrate NADH as electron donor is measured by spectrophotometrically monitoring the consumption of NADH. Under our conditions, 2.5 μM strep-MBP-ROXY9 or 0.2 μM strep-MBP-GRXC2 rescued about 6% (0.018 μM) of the GAPDH activity that could be recovered with DTT (approximately 0.3 μM;

Supplementary Fig. 10), while GSH alone rescued roughly 1.5% (Fig. 3d). The exchange of the CCLC motif against CPYC or CCLS did not alter the activity of strep-MBP-ROXY9 (Fig. 3d). Activation of GAPDH required GSH, providing evidence that strep-MBP-ROXY9 can function as a weak GSH-dependent reductase rather than just as an acceptor of glutathione in this assay.

## ROXY9 shows a lower reactivity towards GSSG as compared to GRXC2

To elucidate whether the lack of oxidoreductase activity of ROXY9 on HED, Cys-SG and roGFP2 is caused by a different reactivity towards GSH and/or GSSG as compared to class I GRXs, we subjected strep-MBP-ROXY9 and strep-MBP-GRXC2 to different GSH/GSSG molar ratios at p$H$ 7 and monitored their corresponding redox states using the alkylation shift assay as described above. After incubation of strep-MBP-ROXY9 with GSH and subsequent alkylation, the mobility of the protein was reduced as observed for the DTT-treated sample that was loaded as a reference (Fig. 4a). Similar to what we had observed with the DTT/dithiane redox titration (Fig. 2), increasing oxidative conditions led to the appearance of a protein species with a higher mobility at a redox potential between −220 and −230 mV. This species had the same apparent mobility as the dithiane-treated protein, suggesting that it corresponds to strep-MBP-ROXY9-S$_2$. Starting at a redox potential of approximately −220 mV, an additional protein band with an even higher mobility than that of strep-MBP-ROXY9-S$_2$ appeared (Fig. 4a). When mutating the second cysteine of the C$_{21}$C$_{22}$LC$_{24}$ motif to serine, only the band that corresponds to the C21-C24 disulfide was formed (Fig. 4c). Thus, glutathionylation of Cys22, which has a large exposed surface area (Supplementary Table 1), seems to occur in strep-MBP-ROXY9-S$_2$. Mass spectrometry analysis of the GSSG-treated sample indeed confirmed that the CCLC motif can harbour one intramolecular disulfide plus one glutathionylated cysteine at the same time (Fig. 4b, Supplementary Fig. 11). In order to exclude that the enzymatic inactivity of ROXY9 on roGFP2 in the presence of GSSG is due to a possible inhibitory effect of glutathionylation at Cys22, we repeated the roGFP2 oxidation assay at -200 mV. Again, no oxidase activity was detectable (Supplementary Fig. 12). In the presence of GSSG, Cys49 and Cys61 that flank helix 2 become glutathionylated as well (Supplementary Fig. 13).

Next, we subjected strep-MBP-ROXY9-CCLS, which cannot form the intramolecular disulfide, to different GSH/GSSG ratios (Fig. 4d). Glutathionylation of this protein variant required higher oxidative power than disulfide formation within the wild-type protein. The same alkylation shift pattern was observed with strep-MBP-ROXY9-SCLC (Supplementary Fig. 14) indicating no preferential glutathionylation of either Cys21 or Cys24.

In contrast to strep-MBP-ROXY9, which quantitatively stayed in the reduced form in the presence of GSH, incubation of strep-MBP-GRXC2 with GSH already led to the formation of two oxidized species (Fig. 4e). This can be explained by low amounts (<1%) of GSSG even in freshly prepared GSH preparations[51,52]. One of the protein species had the same apparent mobility as the dithiane-treated sample suggesting that it is strep-MBP-GRXC2-S$_2$. The other oxidized form with a lower mobility most likely represents a glutathionylated protein species (strep-MBP-GRXC2-SG). Mass spectrometric analysis of the GSH-treated sample confirmed the existence of a glutathionylated peptide (Fig. 4f, Supplementary Fig. S15). However, the intramolecular disulfide-containing peptide was under-represented when compared to the results of the non-reducing SDS PAGE analysis. This might be due to unfavourable ionisation properties of the disulfide-containing peptide or the disulfide might negatively affect tryptic cleavage of the respective peptide before mass spectrometric analysis. Strikingly, the ratio of the two oxidized GRXC2 protein species remained fairly constant over a wide range of redox potentials. As already observed before (Fig. 2c), strep-MBP-GRXC2-S$_2$ became the predominant protein

species in the presence of GSSG. This higher level of oxidation was confirmed by mass spectrometry (Fig. 4f). Under these conditions, we observed some glutathionylation of Cys105 at the beginning of helix 3 (Supplementary Fig. 16). Apparently, even low levels of GSSG in the freshly prepared GSH result in both glutathionylation and intramolecular disulfide formation. The different reactivities of strep-MBP-ROXY9 and strep-MBP-GRXC2 with GSSG were reproduced with His-ROXY9 and His-GRXC2 (Supplementary Fig. 17)

In order to test, whether oxidation of the active site cysteines over a wide range of GSH/GSSG redox potentials might be common for class I GRXs, *Homo sapiens* GRX1 (Gene ID: 2745) as a prototypical representative was prepared as a strep-MBP-fusion protein. Surprisingly, the expected intramolecular disulfide between the two cysteines of the CPYC motif was barely formed in the presence of dithiane (Supplementary Fig. 18). However, similar to what we observed for strep-MBP-GRXC2, strep-MBP-HsGRX1 became oxidized already upon addition of GSH, resulting in two main protein species which persisted over a wide range of GSH/GSSG ratios (Fig. 4g). The species with the highest mobility had the same mobility as the weak band appearing after dithiane treatment and most likely corresponds to strep-MBP-HsGRX1-S$_2$. The less oxidized species is likely to be strep-MBP-HsGRX1-SG. Similar to what we observed for strep-MBP-GRXC2, GSSG treatment led to lower amounts of strep-MBP-HsGRX1-SG and higher amounts of strep-MBP-HsGRX1-S$_2$ and additional oxidations of other cysteines.

Since the redox states of strep-MBP-ROXY9 and strep-MBP-GRXC2 were different in the presence of GSH, we wondered whether the unique CCLC signature was responsible for this feature. However, a strep-MBP-ROXY9-CPYC variant also stayed reduced in the presence of freshly prepared GSH (Supplementary Fig. 19).

Next, we tested whether strep-MBP-ROXY9-S$_2$ and strep-MBP-GRXC2-S$_2$ can be reduced by GSH (Fig. 5). To this aim, both proteins were first oxidized with dithiane to yield the protein species with an intramolecular disulfide. Subsequently, proteins were desalted and incubated in the presence of either DTT, GSH or GSH in combination with NADPH and GR. Strep-MBP-ROXY9 was reduced under all three conditions with GSH alone being somewhat less efficient than DTT or GSH/NADPH/GR. Strep-MBP-GRXC2 was quantitatively reduced by DTT and GSH/NADPH/GR. However, in the presence of GSH alone, around half of the proteins in the sample retained the intramolecular disulfide, whereas the other half was reduced to the glutathionylated species. Thus, independent of whether the experiments were started with reduced or oxidized proteins, most of strep-MBP-ROXY9 was in the reduced state in the presence of GSH with spurious amounts of GSSG, while strep-MBP-GRXC2 became oxidized, with about half of the proteins in the sample being glutathionylated and half containing the intramolecular disulfide.

## The p$Ka$ of the putative catalytic cysteine is lower in GRXC2 than in ROXY9

To investigate the potential reactivity of Cys$_A$ of ROXY9 and GRXC2 as a nucleophile under physiological p$H$, we compared the apparent p$K_a$ values of the cysteines of strep-MBP-ROXY9 and strep-MBP-GRXC2. Proteins were incubated at different p$H$ values with iodoacetamide (IAM), which reacts only with cysteines in their deprotonated thiolate form and protects them from subsequent alkylation by mmPEG. IAM-labelled proteins were TCA-precipitated and subjected to the mmPEG alkylation shift assay (Fig. 6). In a subfraction of the protein, two of the five cysteines in strep-MBP-ROXY9 were not labelled with mmPEG at p$H$ 3.0 and p$H$ 3.5. This might be due to formation of an intramolecular disulfide initiated by the nucleophilic attack of deprotonated Cys$_A$ on Cys$_B$, with molecular oxygen serving as an electron acceptor. An increasing reactivity towards IAM due to deprotonation of one thiol set in between p$H$ 4 and 4.5 (Fig. 6a). p$K_a$ values in this range have been reported for Cys$_A$ of several class I GRXs, although they can be as low as 2.8[51]. The other cysteines had higher p$K_a$ values and at p$H$ 8, all

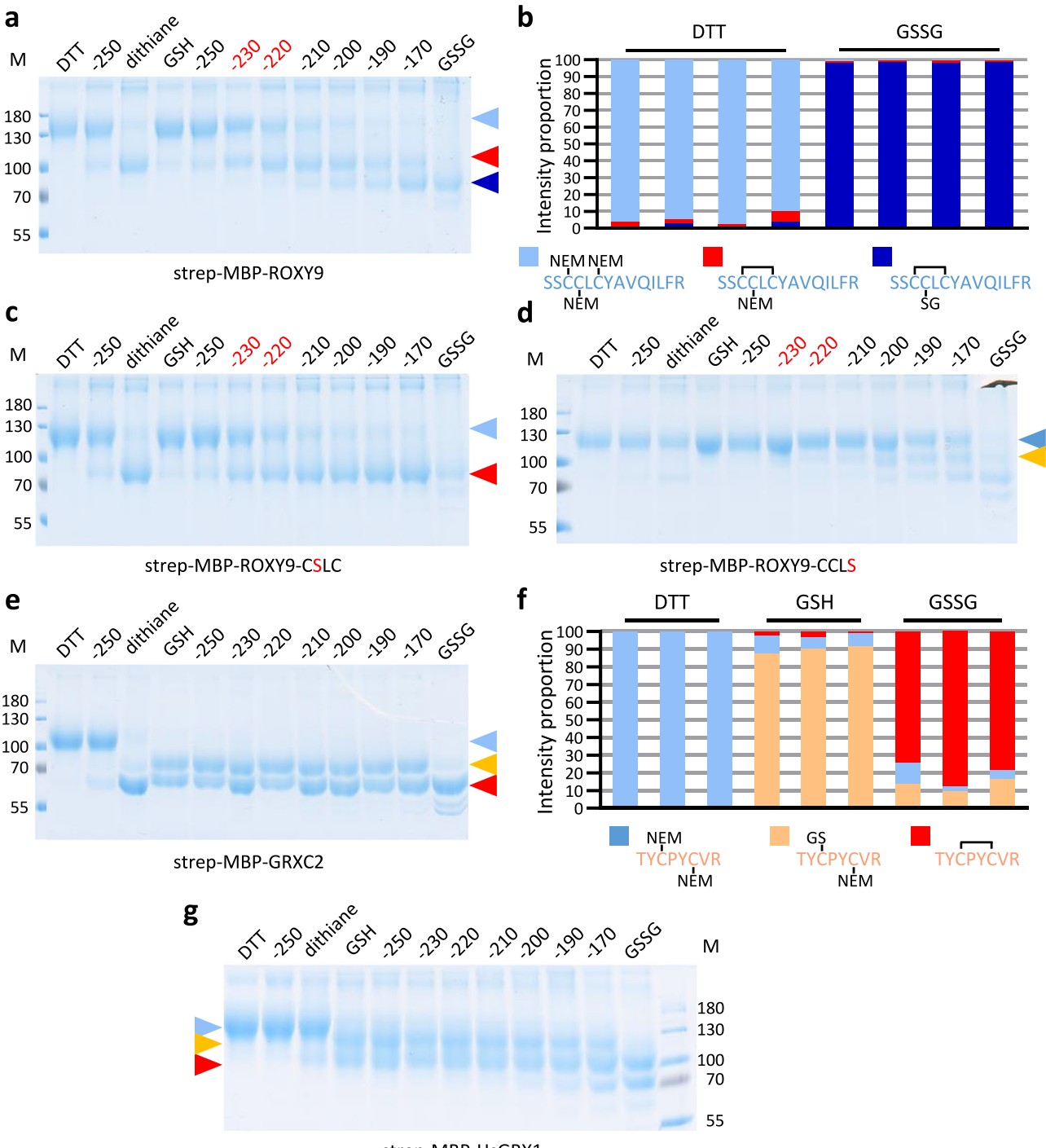

**Fig. 4 | Strep-MBP-GRCX2 but not strep-MBP-ROXY9 is stably glutathionylated at CysA.** Strep-MBP-ROXY9 and its active site variants (CSLC, CCLS), strep-MBP-GRXC2 and strep-MBP-HsGRX1 (approximately 0.2 mM) were pre-reduced with DTT, desalted and mixed with DTT/dithiane (10 mM) and GSH/GSSG (10 mM) buffers at the redox potentials as calculated by the Nernst equation, TCA-precipitated, alkylated with mmPEG for analysis on a non-reducing SDS PAGE (**a, c, d, e, g**) or with NEM for mass spectrometry (**b, f**). The midpoint redox potentials (mV) of strep-MBP-ROXY9 and its derivatives are indicated in red. The colour code of the triangles corresponds to the colour code of the redox state as determined by mass spectrometry. Molecular masses of marker proteins (M) are indicated in kDa. (**b, f**) Relative intensity proportions of peptides containing the active site with the indicated modifications. The results are from three or four replicates, with each replicate representing an independent treatment. Source data are provided as a Source Data file.

cysteines were found to be deprotonated. When mutating all but the putative catalytic cysteine (CysA) to serines, the remaining Cys21 had a $pK_a$ between 6 and 7 (Fig. 6b). Obviously, the mutation of other cysteines to serines can shift the $pK_a$ of the catalytic cysteine of strep-MBP-ROXY9 to higher values. This effect has been observed for pig GRX[44], which has a catalytic cysteine (Cys22) with a $pK_a$ of 3.8. Changing Cys25 to either serine or alanine increased its $pK_a$ to 4.9 and 5.9, respectively. The $pK_a$ of CysA of strep-MBP-GRXC2 was between 3.0 and 3.5 in the wild-type protein and between 3.5 and 4.0 in the strep-MBP-GRXC2-CSS variant (Fig. 6c, d).

## The loop structure C-terminal to helix 2 is predicted to be different for ROXY9 and GRXC2

To evaluate structural differences between ROXY9 and GRXC2, we compared the respective predictions according to AlphaFold (Fig. 7). The conserved lysine residue N-terminal to the active site, Cys$_A$, helix 1 and helix 3 as well as the ß-sheets of the thioredoxin fold are almost perfectly superimposed. In contrast, the position of helix 2, which is by two amino acids shorter in ROXY9, is slightly shifted. Conspicuously, the loop adjacent to helix 2, which ends with a conserved VP motif, points towards helix 2 in GRXC2 but not in ROXY9. The structures become superimposable again at the VP sequence, albeit not perfectly. These differences as well as alterations in conserved amino acids known or predicted to contribute to a functional scaffold site (Gln/Glu

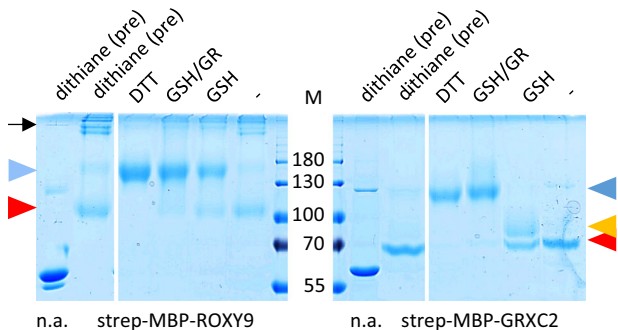

**Fig. 5 | Oxidized strep-MBP-ROXY9 but not oxidized strep-MBP-GRXC2 can be reduced by freshly prepared GSH.** Proteins were first reduced with DTT, desalted and subsequently treated with dithiane (pre). After a second desalting step, they were either treated with 10 mM DTT, 10 mM GSH and glutathione reductase (GR) or 10 mM GSH or were left untreated (-). After TCA precipitation, reduced cysteines were labelled with 5 kDa mmPEG. Samples were separated by non-reducing SDS PAGE. The first lanes contain the dithiane-treated proteins without alkylation (n.a.). The blue triangle denotes the reduced protein, the red triangle the disulfide-containing protein, the ocher yellow triangle the glutathionylated protein. The black arrow denotes DTT-sensitive oligomers. Molecular masses of marker proteins (M) are indicated in kDa. All samples were run on the same gel. The uncropped gel is provided as a Source Data file.

in helix 2 and Arg/Thr in the loop)[13] might be the reason why GSSG is not bound in a way that stable glutathionylation at low GSSG/GSH ratios is favoured. The amino acids which contact the carboxyl group of the glutamate moiety of GSH are located C-terminally to the conserved GG sequence at the beginning of helix 3. An aspartate is often found in class I GRXs while an asparagine is found in ROXY9. Due to different hydrogen bonding even within class I GRXs[53], it is difficult to predict whether the different sequence in this region of ROXY9 contributes to positioning GSH in an unfavourable way with respect to oxidoreductase activity.

## Discussion

GRXs have been grouped into three classes according to their active site sequences, which are either variants of the CPYC/S motif (class I), the highly conserved CGFS sequence (class II) or variants of the CCMC/S motif (class III)[21]. In general, class I GRXs are enzymatically active as oxidoreductases on small disulfide model substrates like HED or Cys-SG, which functionally discriminates them from class II GRXs[14,15]. Here, we show that class III GRX ROXY9 has no measurable oxidoreductase activity on substrates HED, Cys-SG or roGFP2 but that it shows weak GSH-dependent reductase activity on GAPDH-SG. ROXY9 is oxidized to ROXY9-S$_2$ depending on the redox potential of the GSH/GSSG couple, while class I GRXs adopt a stable equilibrium between the glutathionylated and the S$_2$-containing state over a wide range of redox potentials established by GSH/GSSG. We thus explain the enzymatic inactivity of ROXY9 by a poor scaffold site for standard glutathionylated substrates.

Analysis of class I and class II GRXs from different organisms have provided an extensive data set on their biochemical and biophysical characteristics. Biochemical characterization of class III GRXs from plants has been precluded by problems encountered when attempting to purify soluble recombinant proteins. Couturier et al. (2010) obtained small amounts of poplar PtGRXS7.2 using an *E.coli* expression system. The protein exhibited weak oxidoreductase activity in the HED assay. The UV/Visible absorption spectrum was suggestive of the presence of a [2Fe−2S] cluster. Chimeric class I PtGRXs containing the class III GRX-specific SCCMC motif instead of the YCGYC (PtGRXC1) or YCPYC (PtGRXC4) motifs, respectively, showed moderately to severely

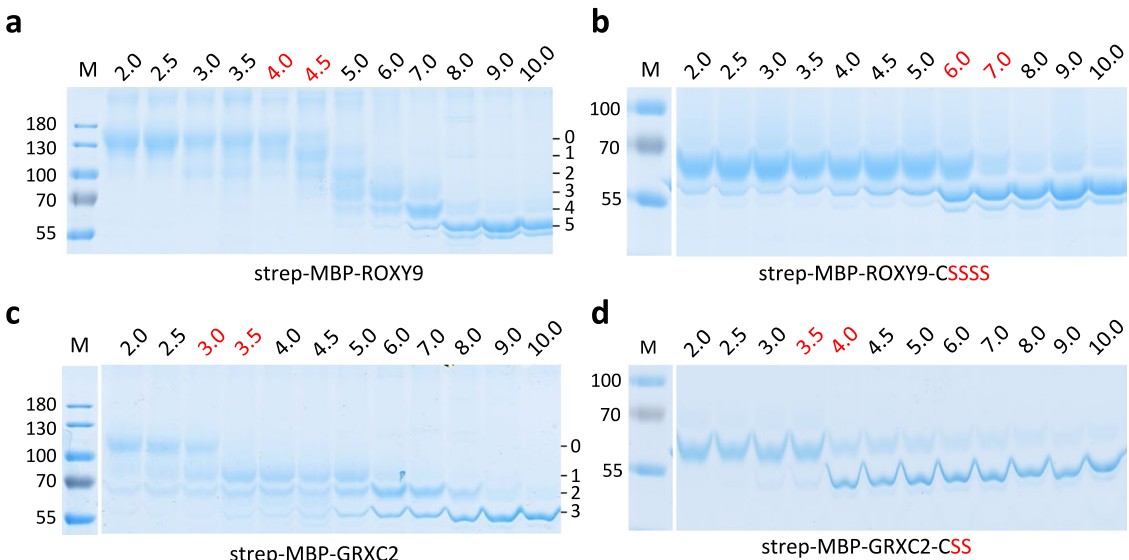

**Fig. 6 | The p$Ka$ of the most reactive cysteine is lower in strep-MBP-GRXC2 than in strep-MBP-ROXY9.** Strep-MBP-ROXY9 (**a**), strep-MBP-GRXC2 (**c**) and their variants encoding only a cysteine at the beginning of helix 1 (**b, d**) were reduced with 10 mM DTT, desalted and incubated with 100 mM iodoacetamide (IAM) at different pH values as indicated. Samples were subsequently TCA-precipitated, labelled with 5 kDa mmPEG and separated by non-reducing SDS PAGE (10%). Red numbers denote the pH at which the transition from the protonated to the de-protonated state of the most reactive thiol occurs. Numbers on the right indicate the numbers of de-protonated (IAM-labelled) thiols. Numbers adjacent to lanes M indicate the molecular mass of marker proteins in kDa. Uncropped gels are provided in the Source Data file.

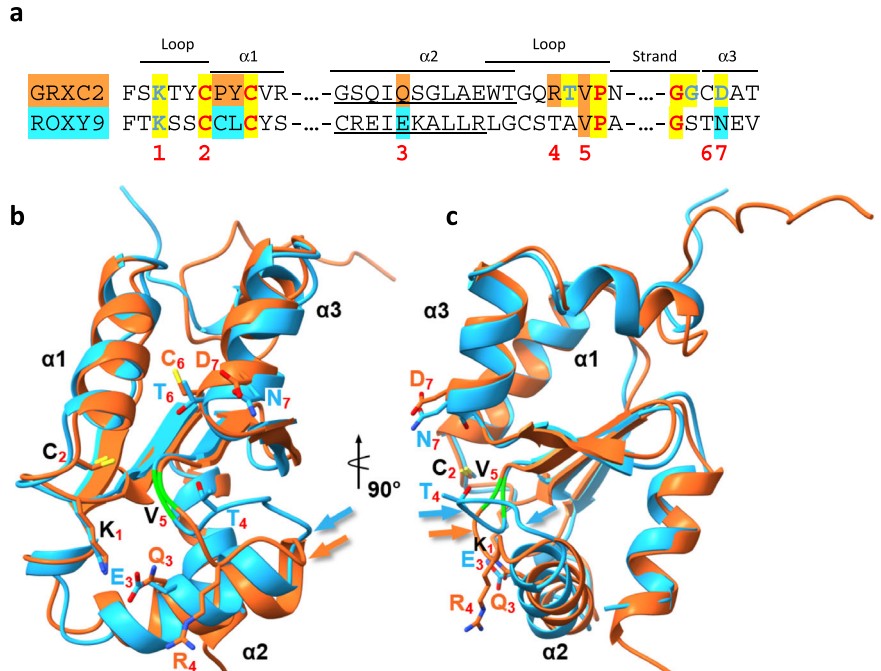

**Fig. 7 | Structural analysis of ROXY9 and GRXC2. a** Primary structure comparison of GRXC2 and ROXY9. Red numbers denote amino acids that may contact GSH. Colours indicate different degrees of sequence conservation as in Fig. 1b. Amino acids forming α helix 2 of the thioredoxin fold according to AlphaFold are underlined. **b,c** Super-positioning of the Alphafold-generated structures of ROXY9 (cyan) and GRXC2 (orange). Side chains of amino acids highlighted by red numbers in (**a**) are shown. The backbone of the valine (5) is shown in green, the arrows points at the loop structures that deviate between the two proteins. In the right panel, the molecule was rotated by 90° to better visualize the relative positions of α helices 2 and the valine backbones. Structures are found on https://alphafold.ebi.ac.uk/entry/O82255 (ROXY9) and https://alphafold.ebi.ac.uk/entry/B3H604 (GRXC2).

reduced oxidoreductase activity depending on the construct, allowing the conclusion that the peculiar active sites of class III GRXs are principally permissive for oxidoreductase activity. Xu et al. (2021) detected no HED activity for 22 recombinant poplar class III GRXs, including PtGRXS7.2 which they renamed PtGRXA7. Instead, they found cumene hydroperoxidase activity for 14 PtGRXs. However, it has to be noted that the reported specific activity of poplar GRXs was in the order of 0.1 μmol/s per μmol protein, which is approximately 2000-fold lower as compared to the activity of e.g. yeast CHP peroxidase[54]. By applying the baculovirus expression system in insect cells, we succeeded in obtaining milligram amounts of ROXY9, a class III GRX from *Arabidopsis thaliana*. No ROXY9 oxidoreductase activity on standard substrates like HED, Cys-SG or roGFP2 was detectable in our hands. This lack of enzymatic activity made us wonder whether recombinant ROXY9 was correctly folded. The following pieces of evidence support the notion that the recombinant protein has adopted a functional thioredoxin fold: (1) The CD spectrum is very similar to that of class I GRXC2 (Fig. 1c); (2) ROXY9 is highly thermostable, which is a typical feature of the thioredoxin fold (Supplementary Fig. 2); (3) an intramolecular disulfide is formed between Cys21 and Cys24 of the $C_{21}CLC_{24}$ signature (Fig. 2); (4) ROXY9 has weak, GSH-dependent reductase activity on GAPDH-SG. We propose that differences in the GSH binding mode compared to that of GRXC2 are responsible for the lack of enzymatic activity.

The most unexpected and intriguing finding of this study concerns the redox states of ROXY9 and class I GRXs GRXC2 and HsGRX1 observed in redox titration experiments with either DTT/dithiane or GSSG/GSH at pH 7 (Fig. 1, Fig. 4). In ROXY9, increasing oxidative conditions led to the formation of a disulfide between Cys21 and Cys24 of the $C_{21}CLC_{24}$ signature with midpoint redox potentials at around −230 mV at pH 7.0, independent of whether DTT/dithiane or GSH/GSSG was used. In GRXC2, disulfide formation within the CPYC motif revealed a similar midpoint redox potential between −220 and −230 mV upon

DTT/dithiane treatment, while HsGRX1 was barely oxidized by dithiane (Supplementary Fig. S18). Strikingly, both class I GRXs became oxidized even in freshly prepared GSH buffer, with most of the protein species containing either an intramolecular disulfide or being glutathionylated. Addition of NADPH and GR led to reduction of GRXC2 (Fig. 5) indicating that spurious amounts of GSSG in the GSH solution are responsible for the oxidation. Apparently, glutathionylation of the two class I GRXs by GSSG occurs even at very low GSSG amounts in freshly prepared GSH due to binding of GSSG in a way that favours a trigonal planar bipyramidal transition state[13,15]. After the subsequent thiol exchange reaction leading to glutathionylation at $Cys_A$ and release of one GSH, a nucleophilic attack of the resolving cysteine leads to disulfide formation and the release of the (second) GS⁻ from glutathionylated $Cys_A$. Unexpectedly, the relative ratio of semi-oxidized GRX-SG and fully oxidized GRX-S₂ remained stable over a wide range of GSH/GSSG redox potentials. This might indicate that GS⁻, which is displaced upon intramolecular disulfide formation, stays associated with the scaffold site kinetically facilitating the back reaction towards the glutathionylated state. This scenario might lead to a steady state equilibrium between GRX-SG and GRX-S₂ which is fairly independent of the redox potential of the GSH/GSSG couple. In contrast, glutathionylation of ROXY9 requires higher concentrations of GSSG most likely because of a structurally different scaffold site. The glutathionylated state is not observed on non-reducing PAGE because of rapid thiol exchange reactions either with GSH or the resolving cysteine (Cys24), depending on the GSH/GSSG redox potential. In strep-MBP-ROXY9-CCL<u>S</u>, the catalytic cysteine starts to become stably glutathionylated at redox potentials that are less negative than those for disulfide formation in the wild-type protein (Fig. 4). This is in contrast to class I GRXs from poplar, where glutathionylation occurs at the same redox potential as disulfide formation[33]. This re-enforces the notion that the scaffold site of class I GRXs is suited to facilitate glutathionylation, which distinguishes these GRXs from class III GRX ROXY9. Further evidence for the

hypothesis that glutathionylation is not supported by a suitable scaffold site in ROXY9 comes from the observation that Cys22 of the ROXY9-$C_{21}C_{22}$LS variant started to become glutathionylated at roughly the same redox potential as Cys21, albeit Cys22 is not oriented towards the GSH binding groove (Fig. 4d).

Only little information on the redox states of especially class I GRXs at varying GSH/GSSG ratios is available. Similar to what we observed with AtGRXC2 and HsGRX1, HsGRX1 was reported to become oxidized by GSH at a redox potential of < −250 mV[55]. Consistent with the idea that very low GSSG levels lead to oxidation of class I GRXs is the finding that less than 40% of oxidized E. coli GRX1 can be reduced by GSH[56]. In contrast, oxidised ROXY9 is reduced in the presence of GSH, although the midpoint redox potentials for disulfide formation in the two proteins (−233 mV at pH 7 for E.coli GRX1[36]; between −230 and −240 mV at pH 7 for ROXY9) are very similar.

It has been discussed before, that—due to the differential influence of enzymes and competing reactions in complex cellular systems[4,57]—the midpoint redox potential of GRXs determined in vitro does not necessarily allow predictions on their redox states in vivo. Indeed, according to dynamic imaging with redox-active sensors, the redox potential of the yeast cytosol is in the range of −285 mV[58]. The lowest reported midpoint redox potential of the disulfide of class I GRXs as measured with the DTT/dithiane redox couple is −263 mV for poplar GRXC1[33]. Assuming that the midpoint redox potential of yeast GRXs is not lower than this value, most of the protein should be reduced in vivo. However, 50% of yeast GRXs are oxidized at resting conditions[59], supporting our notion that the midpoint redox potential of the active site disulfide does not determine the redox state of GRXs if they are in an environment with the interacting ligands GSH and GSSG. Consistently, class I poplar GRXS12, which has an active site without a resolving cysteine, is almost 100% glutathionylated when purified from E. coli in the absence of reducing agents[18]. Likewise, a large fraction of Arabidopsis GRXC5 (active site CSYC) is glutathionylated when prepared from E. coli, irrespective of whether GSH is included in the buffers used for purification or not[60]. Thus, even under reducing cellular conditions, GRXS12 and GRXC5 seem to be glutathionylated, or, alternatively, they become readily glutathionylated in the extract. This is consistent with our in vitro data, which provide clear evidence for glutathionylation of class I GRXs at very negative redox potentials. It may be speculated that partial glutathionylation of class I GRXs even under the high GSH/GSSG ratio present in living cells provides a reservoir of glutathionylated GRXs to facilitate rapid glutathionylation of reactive cysteines. At the same time, constant regeneration of reduced GSH by glutathione reductase and NADPH supplies enough reduced class I GRXs with high reactivity for glutathionylated substrates for de-glutathionylation reactions. Finally, the amount of this flexible pool of reduced and glutathionylated GRXs might be independent from local perturbations of the cellular GSH/GSSG redox potential.

It is well established that the interaction of GRXs with glutathione adducts in the so called scaffold site is important for oxidoreductase activity[13–15]. After comparison of the redox titrations of ROXY9 and GRXC2 with either GSH/GSSG or DTT/dithiane, we had concluded that GRXC2, but not ROXY9, accepts low amounts of GSSG as a substrate for an oxidative half-reaction because of its favourable scaffold site leading to glutathionylation at $Cys_A$. Because of a favourable scaffold site for the glutathione moieties of ßME-SG and Cys-SG, $Cys_A$ of GRXs attacks the mixed disulfides, yielding ß-ME or cysteine, respectively, and the GRX-SG intermediate. This is then reduced upon GSH binding to the activator site, which leads to the release of GSSG. We explain the lack of reductase activity of ROXY9 in the HED and the Cys-SG assays by an unfavourable scaffold site as indicated by the redox titrations. Similarly, roGFP2-SG, which is formed upon incubation of roGFP2-$S_2$ with GSH, cannot be reduced by ROXY9.

When acting as oxidases in the presence of GSSG, glutathionylated GRXs transfer their glutathione moiety to reactive thiols of their substrates. In ROXY9, the glutathionylated protein species is very short-lived, which explains the lack of oxidase activity of ROXY9 on roGFP2. Strep-MBP-ROXY9-CCLS, which lacks the resolving cysteine, became glutathionylated most likely at both cysteines of the CCLS signature upon GSSG treatment (Fig. 4d). Due to the poor scaffold site, glutathione might not be positioned in a way that favours the disulfide exchange reaction with the nucleophilic cysteine of the roGFP2 substrate. This is different from several class II GRXs, which have at least a weak oxidase activity on roGFP2[13,14,50].

Still, we discovered weak GSH-dependent reductase activity with GAPDH-SG. In our assay, 2.5 µM strep-MBP-ROXY9 deglutathionylated 0.018 µM GAPDH in 20 min. At this ratio of enzyme over substrate, the reaction should not require regeneration of the enzyme by GSH. The requirement for GSH may be explained by assuming that ROXY9 weakly binds GSH in the scaffold site, leading to its deprotonation and thus to its activation[47,61]. This activated GS⁻ might attack the glutathione mixed disulfide of GAPDH-SG, leading to GAPDH and GSSG. Likewise, GRXC2, which might cycle between GRXC2-SG and GRXC2-$S_2$ in the absence of GR and NADPH, might provide activated GS⁻. Given the larger effect of GR in combination with NADPH on the redox state of GRXC2 as compared to the redox state of ROXY9 (Fig. 5), we assume that addition of these compounds to the assay[6] would increase the difference between the de-glutathionylation activities of both proteins. In contrast, GAPDH deglutathionylation activity mediated by class II C. reinhardtti GRX3 is independent from GSH when GRX3 is maintained in the reduced state by light-driven electron transport through photosystem I.

The observed weak or undetectable oxidoreductase activity of ROXY9 was not necessarily expected from the primary sequence, since ROXY9 rather resembles class I than class II GRXs. Importantly, ROXY9 lacks the oxidoreductase-inhibitory five amino acid loop between the highly conserved lysine residue and the active site cysteine (Fig. 1b). Removal of this loop in class II GRX HsGRX5 had generated an enzyme variant that—in contrast to the wild-type—had detectable reductase activity with roGFP2 (approximately 1.5% of class I HsGRX2) and HED (approximately 0.5% of class I HsGRX2). With an additional change of the CGFS signature to CSYC, reduction of roGFP2 increased only slightly (approximately to 1.8% of class I HsGRX2), but HED activity was increased more efficiently (approximately to 2.5% of class I HsGRX2)[14]. A weak enzymatic activity of class II HsGRX5 after deletion of the loop was also reported in an independent study which used Cys-SG as a substrate[15]. Even when we replaced the ROXY9 CCLC motif by the class I motif CPYC, no oxidoreductase activities could be detected, neither with HED, Cys-SG nor with roGFP2 (Fig. 3). Thus, other structural differences between ROXY9 and class I GRXs must be responsible for the observation that we did not even regain minimal oxidoreductase activity with the strep-MBP-ROXY9-CPYC variant.

The largest predicted structural differences between GRXC2 and ROXY9 are observed for helix 2 of the thioredoxin fold and the adjacent loop that clearly takes a different path in ROXY9 (Fig. 7). Most class I GRXs contain a conserved glutamine in helix 2, which contacts the carboxyl group of the glycine residue of the non-covalently bound GSH ligand within an FeS containing holoprotein (PtGRXC1: PDB(2E7P); HsGRXC2: PDB(2HPT9); ScGRX6: PDB(3L4N)). However, these interactions are not visible when GSH is bound independently of the FeS cluster (HsGRX1: PDB(1B4Q); EcGRX2: PDB(7DKP)). In class II GRXs, an arginine or lysine is often found at this position, which can also contact the carboxyl group of the ligand (e.g. HsGRX5: PDB (2WUL)). In contrast, ROXY9 encodes a glutamate, the negative charge of which may even lead to repulsion of the ligand.

Within the loop region C-terminal to helix 2, most class I GRXs contain a basic residue (Fig. 1b). In HsGRXC2 (PDB(2HPT9)), this arginine pulls the loop towards helix 2 through interactions with a

glutamate residue. In ScGRX6 (PDB(3L4N)), the arginine entertains interactions with amino acids from different parts of the thioredoxin fold and a water-bridged interaction with the amide group of the GSH-encoded glycine. In EcGRX2, the arginine contacts the carboxyl group of the glycine moiety of GSH (PDB(7DKP)). Mutation of the corresponding Arg153 in ScGRX7 confirmed its critical role for the scaffold site as revealed by the 60% reduction of the catalytic efficiency on the substrate Cys-SG[15]. Class II GRXs have a WP motif in this position (Fig. 1b) which is critically required for FeS cluster binding[14]. The lack of this basic residue in ROXY9 might add to its poor scaffold site either through its effect on the position of the loop or on other interactions within the thioredoxin fold.

The end of the loop contains a valine within a highly conserved TVP signature in class I GRXs. It is known, that the amide- and carbonyl groups of the valine backbone of the VP motif form one or two backbone-backbone hydrogen bonds with the cysteine of bound GSH. Thus, this region appears to be critical with respect to positioning of the cysteine residue of GSH within the GSH binding groove. The valine backbones of GRXC2 and ROXY9 cannot be perfectly superimposed due to the different positioning of the loop in the two proteins. This feature might also contribute to the poor scaffold site of ROXY9. It remains to be investigated whether changing the amino acid sequence of ROXY9 into the sequence of class I GRXs at the discussed positions will be sufficient for a gain of function with respect to oxidoreductase activity, or whether other structural perturbations in the protein are responsible for its dysfunctionalized oxidoreductase activity.

Interestingly, the loop region and even the valine itself are weakly conserved in other ROXYs, with e.g. ROXY1 having a longer loop and ROXY20 having a shorter loop (Fig. 1b). Thus, this region and its potential impact on the exact positioning of GSH might not have been under evolutionary pressure in class III GRXs. Still, other amino acids involved in GSH binding, i.e. the lysine residue preceding the active site, which forms electrostatic interactions with the carboxyl group of the glycine in GSH, the active site cysteines and polar amino acids interacting with the carboxyl group of the glutamate in GSH are conserved in most ROXYs, suggesting that GSH binding might be required for activity. We thus conclude, that ROXY9 and maybe other class III GRXs will not efficiently take part in reactions on glutathione mixed disulfides unless the GSH scaffold site is altered by an induced fit of a specific substrate. The resulting poor overlap with the functions of class I GRXs might be advantageous in evolutionary terms.

*In planta*, class III GRXs regulate the activity of TGA transcription factors through yet unknown mechanisms[27–29]. There is recent evidence that maize class III GRX MSCA1 controls the formation of a disulfide-bridged dimer of TGA transcription factor FEA4 to regulate its DNA binding activity and finally meristem size[28]. A FEA4-MSCA1 mixed disulfde was detected on non-reducing SDS gels. Since oxidation via a mixed disulfide does not require binding of GSH to the scaffold site, it is plausible that some class III GRXs might act through this mechanism. ROXY9 controls the activity of TGA1[29]. In contrast to the MSCA1-FEA4 interaction, which depends on the active site cysteines of MSCA1 and a redox-active cysteine of FEA4[28], the ROXY9-TGA1 interaction is independent from the CCLC signature[62]. It remains to be shown whether ROXY9 controls TGA1 activity through a direct redox mechanism or whether the highly solvent-exposed CCLC signature is important for redox-independent protein-protein interactions within regulatory complexes.

## Methods
### Purification of recombinant GRXs
The open reading frames (ORFs) of ROXY9 and its variants, GRXC2, GRXC2-CSS and HsGRX1 were amplified from plasmids described before (ROXY9[62]; GRXC2[63]; HsGRX1[64]). Primers contained extensions for ligation-independent cloning (Supplementary Table 2). PCR-amplified ORFs were inserted into a variant of vector 438-C of the

438 series of MacroLab vectors (modified pFastBac, Addgene plasmid # 48296)[65]. In this variant, the His-tag was exchanged against the Strep-tag® II (IBA Göttingen, Germany). ROXY9 and GRXC2 sequences were also cloned into the original 438-C.

Production of bacmids and baculoviruses was done essentially as described[66]. Protein expression was induced in *Triplusia ni* Hi5 cells by infecting a 600 mL starting culture ($1.0 \times 10^6$ cells/mL) in a 3 l flask with 100 to 400 µL of V1 viruses, depending on its quality and age. On day 2, typically a concentration of around $2.0 \times 10^6$ cells/mL was measured. The culture was split into two 300 ml cultures and each flask was diluted to $1.0 \times 10^6$ cells/mL, usually leading to a final volume of 600 mL that complies with the ratio of air to cell medium of 4:1. On day 3, the Hi5 cells typically did not further divide and started to increase in size. Cells were harvested when the percentage of viable cells as measured in a Casy cell counter (OLS OMNI Life Science) dropped below 88%, usually on day 4 or 5. The cell suspension from each flask was centrifuged at 24 000x g for 30 min at 4 °C. The pellet was resuspended in 12 mL lysis buffer (25 mM HEPES pH 7.5, 100 mM NaCl, 5% (w/v) glycerol, 5 mM GSH). The two suspensions obtained from one culture flask each were combined in a 50 mL falcon tube, frozen in liquid nitrogen and stored at −70 °C. For the experiments shown in Supplementary Figs. 5 and 7a, 1 mM DTT was used. For further processing, the frozen suspensions were thawed in a beaker containing cold water in the cold room. Cell lysis was performed by sonication for 5 min in ice with alternating pulses of 0.4 s with intervals of 0.4 s (Sonoplus sonifier/MS73 sonotrode). The soluble and insoluble fractions were separated by centrifugation for 45 min at 48,000× g. The supernatant was filtered through a stack of one 1.2 µm syringe filter connected to one 0.45 µm syringe filter. The cleared lysate was applied to either a pre-equilibrated 5 ml MBPTrap or HisTrap column (GE Healthcare Biosciences) connected to an Äkta_primeplus device at a flow rate of 1 ml/min. After absorbance returned to baseline, the bound protein was eluted from the column with 100 mM maltose or 250 mM imidazole in lysis buffer. The fractions containing Strep-MBP-GRX or His-GRX were combined and stored at 4°C. In general, experiments with the purified protein were performed with protein batches that had been stored for no more than 2-3 days. In order to change the GSH in the lysis buffer for other redox-active compounds, proteins were desalted with a MiniTrap column. In case the protein was purified in the presence of DTT, buffer exchange was done by dialysis.

For experiments with untagged proteins, samples were treated overnight with 6xHis-TEV protease (Invitrogen) according to the manufacturer's instructions. The cleaved protein was loaded onto a Superdex 200 (10/300 GL) column connected to an ÄKTA_pure device and eluted with lysis buffer. Fractions were analyzed by SDS PAGE and those identified to contain tag-free GRXs were combined and subsequently spin-concentrated with a VivaSpin MWCO 3k concentrator at 4000 rpm and 4 °C. All buffers were degassed before use.

### Circular dichroism spectroscopy
To analyse the secondary structure content and thermal stability of ROXY9 and GRXC2, far-UV spectra and thermal unfolding data were collected using circular dichroism spectroscopy (Chirascan, Applied Photophysics). A MiniTrap desalting column was used to change the buffer of the protein samples to 20 mM $Na_2HPO_4$, pH 7.5, 50 mM NaF, 5% (w/v) glycerol containing either 1 mM DTT or 0.5 mM GSH as reducing agents. Far-UV spectra were collected in the range of 180–260 nm with a step size of 1 nm and at least 20 accumulations for 0.5 s per wavelength. Due to the poor extinction coefficient at 280 nm, determination of protein concentration at the concentrations employed proved inaccurate. Instead, we normalized the data utilizing the absorbance at 220 nm obtained in parallel to circular dichroism data. All spectra were corrected for the background intrinsic to the reducing agent, and averaged after buffer correction. Unfolding data were collected at 222 nm from 20–95 °C (real sample temperature was

determined using a temperature probe) with a ramping speed of 1 °C min$^{-1}$. At each temperature, data were averaged over 10 s.

## Determination of free thiols

The redox state of GRXs was determined by alkylation of free thiols and subsequent analysis of the mobility of the alkylated protein on non-reducing SDS PAGE[34]. For the analysis of 13 redox conditions, around 75 µg of the respective strep-MBP-GRX proteins were reduced using 10 mM DTT in 150 µl of 100 mM HEPES p$H$ 7.0. After incubation for at least 1 h, the DTT was removed by loading the sample onto the bead bed of a MiniTrap desalting column pre-equilibrated with 100 mM HEPES p$H$ 7.0. After the solution had entered the column, 550 µl of 100 mM HEPES p$H$ 7.0 were loaded on the column and 300 µl of 100 mM HEPES p$H$ 7.0 were used to elute the protein. Subsequently, 20 µl (corresponding roughly to 2.5 µg) of the desalted protein was mixed with 30 µl of 100 mM HEPES p$H$ 7.0 buffer containing different redox agents at a final concentration of 10 mM. Samples were incubated for 2 h at room temperature, followed by the addition of 50 µl 20% (v/v) trichloroacetic acid (TCA). Samples were either stored at -20°C or directly processed by centrifugation at 11200 g for 30 min at 4°C. The supernatant was removed and the pellet was washed by adding 800 µl pre-cooled (-20°C) acetone. The pellets were dried and dissolved in 15 µl alkylation buffer (100 mM Tris/HCl p$H$ 8.0, 1% (w/v) SDS) with or without 5 mM methoxyl-PEG-maleimide (mmPEG) of 5 kDa. The samples were alkylated for 45 min at room temperature, mixed with 15 µL non-reducing Laemmli buffer (4% SDS, 120 mM Tris/HCl p$H$ 6.8, 20% glycerol, 0.02% bromophenol blue) and subjected to 12% SDS PAGE. Gels were stained with Coomassie Brilliant Blue. For mass spectrometry analysis, three to four independent treatments of one protein preparation were performed. The starting protein concentration was doubled and 5 µg was loaded per lane. Alkylation was done in 5 mM N-ethylmaleimide (NEM) instead of mmPEG. After brief staining with Coomassie Brilliant Blue, the bands were excised and stored in 10% acetic acid.

For p$H$ titration assays[34], different buffers with a p$H$ from 2.0 to 10.0 were prepared at a normality of 0.1. A citrate buffer system was used for p$H$ 2.0 to 6.0, $KH_2PO_4$/$Na_2HPO_4$ for p$H$ 6.5 to 8.0, and $Na_2B_4O_7$ for pH 8.5 to 10.0. Protein samples were pre-reduced as described above, but 150 µg protein per 150 µl were used. The mixture was desalted against 5 mM HEPES p$H$ 7.0. Subsequently, 10 µl (corresponding roughly to 2.5 µg) of the desalted protein was mixed with 40 µl of the respective p$H$ buffer and 5 µl 100 mM iodoacetamide (IAM). After 1 h incubation time, 55 µL 20% (v/v) TCA were added and the mmPEG labelling was carried out as described above.

## HED/Cys-SG and CHP assays

If not otherwise indicated, strep-MBP-GRX samples were reduced with 10 mM DTT for 2 hours and the buffer was exchanged against 100 mM Tris/HCl p$H$ 8.0, 1 mM EDTA[15] using PD MiniTrap G-25 desalting columns (GE Healthcare). The final reaction mixtures (200 µl) contained 100 mM Tris/HCl p$H$ 8.0, 0.2 mM NADPH, 1 mM GSH (40 µM for the Cys-SG assay), 0.3 U/µl glutathione reductase (GR) from *S. cerevisiae* (Sigma-Aldrich), 0.8 mM HED (0.15 mM Cys-SG) and 1.6 µM protein. The reaction mixtures were incubated for 5 min at 25°C before adding the proteins. The consumption of NADPH was measured at 340 nm using a JASCO V-750 spectrophotometer. The linear decline was recorded for 10 s, with the initial absorbance being set to 0. (For CHP assay, see legend in Supplementary Fig. 6).

## roGFP2 reduction/oxidation assay

The coding region of roGFP2[64] was cloned into the *E.coli* expression vector pDEST17, allowing expression roGFP2 fused to a 6xHis-tag. Purification of the protein was done with a HISTrap column connected to the Äkta_prime_plus chromatography. The capacity of recombinant strep-MBP-GRX fusion proteins to oxidize or reduce purified His-

roGFP2 was analysed by ratiometric time-course measurements using a fluorescence plate reader (Synergy HT; BioTek). The redox state of roGFP2 was calculated by measuring the emission at 528 nm after excitation at either 360 nm or 485 nm and calculation of the respective ratio[48]. Reduced or oxidized His-roGFP2 samples were prepared at a final concentration of 2 µM by pre-incubation with 10 mM DTT or 10 mM $H_2O_2$, respectively. Reduced or oxidized His-roGFP2 and reduced strep-MBP-GRX samples were passed through PD MiniTrap G-25 desalting columns (GE Healthcare) so that the proteins were finally in 100 mM Tris/HCl, p$H$ 8.0, 1 mM EDTA.. To test the capability of GRXs to catalyse the reduction of oxidized roGFP2, GSH was added to a final concentration of 2 mM together with 100 µM NADPH and 0.3 U/µl GR from *S. cerevisiae* (Sigma-Aldrich). To test GRX-dependent oxidation of reduced roGFP2, the reaction was started by adding 50 µM GSSG. The final reaction mixtures (100 µl) contained 2 µM His-roGFP2 and 2 µM of the respective strep-MBP-GRX fusion proteins. The reference samples contained either desalted oxidized His-roGFP2 along with 50 µM GSSG and 2 µM strep-MBP-ROXY9 or desalted reduced His-roGFP2[67].

## GAPDH Assay

The GAPDH assay was adapted from Zaffagnini et al. (2008). 6 µM rabbit GAPDH (Sigma-Aldrich) was incubated for 20 min in a total volume of 500 µl of 100 mM Tris/HCl p$H$ 8.0, 1 mM EDTA, 1.4 mM $H_2O_2$ and 3 mM GSH, which led to glutathionylation and inactivation of enzymatic activity. Untreated GAPDH was used as a control. $H_2O_2$ and GSH/GSSG were removed using a PD MiniTrap G-25 desalting column (GE Healthcare). Strep-MBP-GRX samples were reduced with 10 mM DTT for 2 hours and the buffer was exchanged against 100 mM Tris/HCl p$H$ 8.0, 1 mM EDTA using the PD MiniTrap G-25 desalting columns (GE Healthcare). The de-glutathionylation reaction was performed for 20 min in 300 µl containing 0.6 µM GAPDH-SG, 0.2 µM strep-MBP-GRXC2 or 2.5 µM strep MBP-ROXY9 or its derivatives. Each reaction was performed in the absence or presence of 2 mM GSH. Control reactions contained 40 mM DTT. The GAPDH substrate 1,3 bisphosphoglycerate was generated by incubating 13.5 mM 3-phosphoglycerate, 6 mM ATP with 0.012 U/µl 3-phosphoglycerate phosphokinase in 100 mM Tris/HCl p$H$ 8.0, 1 mM EDTA, 15 mM $MgCl_2$ for 20 to 30 min at RT. GAPDH activity was measured after mixing 50 µl of the deglutathionylation reaction with 50 µl of the substrate-generating mixture. The reaction was started by adding 50 µl of 3 mM EDTA and 0.6 mM NADH in 100 mM Tris/HCl p$H$ 8.0. Consumption of NADH was measured every min for one hour using a JASCO V-750 spectrophotometer. The linear decrease in absorbance between 15 and 60 min was fitted to determine GAPDH activity in each sample. All reactions were performed at 25 °C.

## Mass spectrometry

Protein containing polyacrylamide gel slices were prepared for mass spectrometric analysis by 2 × 5 alternating washing steps with 10 mM HEPES, p$H$ 6.8 in water and a 1:1 mixture (v/v) of 100 mM HEPES in water (p$H$ 6.8) and acetonitrile. After drying of gel pieces in a vacuum concentrator, proteins were digested with 80 ng trypsin per gel band in 100 mM HEPES in water (p$H$ 6.8) overnight at 37 °C. Resulting peptides were extracted from the gel piece with an 1:1 mixture (v/v) of 0.1% trifluoroacetic acid and acetonitrile in a ultrasonic water bath for 2 x 15 minutes. After drying of peptides in a vacuum concentrator, peptides were desalted by solid phase extraction (Oasis HLB 96-well µElution plate, Waters) according to the manufacturer's instructions and -100 ng per sample was used for mass spectrometric analysis. Peptides were first separated over 60 minutes on C18 material using an Ultimate 3000 rapid separation system (Thermo Fisher Scientific) as described earlier[68]. Then, tandem mass spectra were record in an online coupled Fusion Lumos Tribrid mass spectrometer (Thermo Fisher Scientific) equipped with a nano-electrospray source and FAIMS

device (carrier gas flow 4.5, compensation voltage −50 mV) in data-dependent positive mode. First, survey spectra were recorded in the orbitrap analyser (resolution 60000, scan range 400-1800, maximum injection time 50 ms, AGC target 100000). Second, 2-10fold charged precursors were selected by the build in quadrupole (isolation window 1.6). Two fragment spectra were recorded in the orbitrap analyser (resolution 30000, scan range auto, maximum injection time 100 ms, AGC target 50000) for each precursor, one after collision-induced-dissociation fragmentation and one after higher-energy collisional dissociation type fragmentation. Cycle time was 2 seconds and already fragment precursors were excluded from fragmentation for the next minute. Protein and peptide identification and quantification was carried out with MaxQuant version 2.2.0.0 (Max-Planck-Institute for Biochemistry, Planegg, Germany) using standard parameters if not stated otherwise. Strep-MBP-ROXY9 and strep-MBP-GRXC2 amino acid sequences were used as templates for database searches as well as common contaminants provided by the build-in contaminant list. Beside methionine oxidation and protein N-terminal acetylation, following cysteine modifications were considered as variable modifications: sulfinic acid, sulfonic acid N-ethylmaleimide, glutathione, disulfide (-H). Additionally, a search for disulfide crosslinks was activated. Modified peptide specific intensities were summed up from the intensities provided in the evidence table form the MaxQuant output from modified peptide variants which where repetitively identified with a reasonable number of peptide spectrum matches.

### Data presentation and reproducibility
GraphPad Prism 10 (GraphPad Software, Inc., San Diego, CA) was used to display the results of the enzymatic assays and the peptide intensities. Detailed results are shown in the Source Data File. Crucial results displayed by non-reduced SDS PAGE were confirmed in different experimental set-ups: Data shown in Fig. 2b, c are reproduced for the redox potential of −250 mV in Fig. 4a, e and are supported by Supplemental Fig. 4c; Data shown in Fig. 4a, 4e were reproduced with proteins with an HA tag as shown in Supplementary Fig. 17 and are supported by Supplementary Fig. 19. Data from Fig. 5 support the results of Fig. 4a.

### Reporting summary
Further information on research design is available in the Nature Portfolio Reporting Summary linked to this article.

## Data availability
Data that support the findings of this study are available within the paper and its supplementary material. The mass spectrometry proteomics data have been deposited to the ProteomeXchange Consortium via the PRIDE [1] partner repository with the dataset identifier PXD042724[69].Structures are found on https://alphafold.ebi.ac.uk/entry/O82255 (ROXY9) and https://alphafold.ebi.ac.uk/entry/B3H604 (GRXC2). Source data are provided with this paper.

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

## Acknowledgements

The work was funded by the Deutsche Forschungsgemeinschaft (DFG; Grant GA330/25 within the DFG priority programme SPP1710 and Grant GA330/27 (P.M.)) and the Dorothea-Schlözer-Program of the Georg-August-University Goettingen (K.Treffon). S.G. was funded by core funding of the Georg-August-University Goettingen. We thank Dr. Guido Kriete (Georg-August-University Goettingen), Dr. Isaac Fianu and Prof. Dr. Patrick Cramer (Max-Planck-Institute for Biophysical Chemistry, Goettingen) for help with the establishment of the baculovirus expression system and Xueyi Jin for help with protein work. We acknowledge support by the Open Access Publication Funds of the Göttingen University.

## Author contributions

P.M. performed the cloning and established the expression system in insect cells and performed the redox titration experiments. S.G. performed the enzymatic assays shown in Fig. 3. The HED assay with unfused ROXY9 was performed by K. Treffon, who also did preliminary work with the GAPDH assay. Mass spectrometry analysis was performed by G.P.; F.v.P and K.Tittmann assisted in gel filtration experiments and CD spectroscopy. C.G. supervised the study and wrote the manuscript together with P.M. All authors discussed the results and gave approval to the final version of the manuscript.

## Funding

## Competing interests

The authors declare no competing interests.
