## [Transparent Peer Review file · Nature Communications]

Molecular basis for the enzymatic inactivity of class III glutaredoxin ROXY9 on standard glutathionylated substrates

Corresponding Author: Professor Christiane Gatz

Version 0:

Reviewer comments:

Reviewer #1

(Remarks to the Author)

The paper entitled "Molecular basis for the enzymatic inactivity of class III glutaredoxins on standard glutathionylated substrates" by Mrozek and collaborators focuses on the biochemical and structural properties of a representative member, ROXY9, of the expanded class III glutaredoxins (GRX) from plants. These GRX usually possess a CCxC signature that differs from the classical CPYC signature present in class I GRX. A comparative analysis of the in vitro redox properties and activity profile of ROXY9 was provided using the class I GRXC2 from Arabidopsis as a reference.

It is shown that, unlike GRXC2, ROXY9 is inactive as a reductase on the small substrate HEDS and as an oxidase on roGFP2. However, both proteins possess the capacity to catalyze the deglutathionylation of GAPDH, although to different extent. Redox titrations revealed the formation of a disulfide bond between the first and the last cysteine of the CCLC signature with a value comprised between -220 to -230 mV, as found in many other GRX. Moreover, the determination of the pKa of the catalytic cysteine indicates that it is not different either from other characterized GRX. The GSSG-mediated glutathionylation of the three other cysteine residues present in ROXY9 is also evidenced but not analyzed in detail. Reduction-Oxidation experiments using DTT (oxidized or reduced), GSH (oxidized, reduced or in mixtures of defined redox potentials) point to differences between ROXY9 and AtGRXC2. It is observed that GRXC2 is oxidized to roughly equal amounts of a glutathionylated form (GRX-SG) and a form with an intramolecular disulfide (GRX-S2) over a wide range of GSH/GSSG ratios whereas ROXY9 readily forms an intramolecular disulfide and glutathionylation of the first reactive cysteine was not observed, suggesting that its formation is transient. From structural modeling, it is hypothesized that this different reactivity would be due to subtle differences in the GSH binding mode and that this feature might have evolved to avoid overlapping functions with class I GRXs in plants.

Overall, this study focuses on an important and highly relevant question that deserves to be explored since the analysis of the oxidoreductase activity of these specific GRX was hampered in most cases in the past by the inability to express recombinant proteins. There are substantial amounts of experiments supporting the conclusions. While most experiments have been well achieved and the results discussed with respect to the known literature, I have quite a number of concerns, comments and suggestions for some of them.

Introduction

- lines 70-72: "When functioning as a reductase of glutathionylated substrates, the GSH moiety of the substrate has to be positioned into the GRX GSH binding groove so that the thiol points towards CysA."

In this context, glutathione forms a disulfide with a protein cysteine residue. So, I would avoid (i) using GSH (here and at any places where it does not refer to reduced glutathione) which is used for the reduced form of glutathione and (ii) saying that the thiol points towards CysA since it is not in the thiol form.

-Line 94: remove the f after the references 13 and 14.

- Lines 94-96 : "Class III GRXs do not encode the class II-specific five amino acid loop which interferes with oxidoreductase activity nor the proline in the active site which might interfere with FeS cluster assembly."

Please provide references again for the first aspect. For the question of the proline, I assume that this refers to class I GRX and notably the work performed with GRX from poplar (Rouhier et al 2007, doi: 10.1073/pnas.0702268104).

Results

1. It is interesting that a class III GRX can be produced in an insect cell system, even though it is fused to a strep-MBP tag. I have several comments here. It would be informative to know whether the protein is expressed with an Fe-S cluster, and whether it is purified in a reduced or oxidized form.

Indeed, it is mentioned that a gel filtration performed with DTT indicates a monomeric state, but the reader would be tempted

to conclude that DTT may be present to avoid oligomerization.

Also, it is shown that it is possible to remove the tag and keep a soluble protein. But there is no mention of the oligomeric state and stability of the protein. In fact, it has to be justified why such a bulky tagged protein is used throughout the study, because using an untagged protein would be much more relevant.

2. The results of the CD analysis (Figure c) puzzle me, since it is indicated that adding GSH increases the secondary structure content. It is rather unexpected if we refer to known GRX structures that were solved with or without bound glutathione and for which there are no changes visible. Could it be that the effect is due to the strepMBP tag? In principle, the changes in the secondary structure content can be quantified.

3. Concerning the experiments about the thermal stability, it would be informative to examine the redox state of the proteins before analysis and compare reduced and oxidized forms and even as-purified proteins.

4. Alkylation experiments using 5kDa mmPEG have been used extensively here to evaluate the redox properties of the proteins. However, the retarded migration observed on SDS PAGE gel is much larger than expected (differences of approx. 70 kDa instead of 25 kDa). This has to be at least discussed. Are there cysteine residues in the tag?

Evaluation of the effect of dithiane on the proteins, demonstrating the oxidation of only the two cysteine residues forming an intramolecular disulfide on ROXY9 or GRXC2, has been shown only in fig. 5, while I would have expected to see the results prior other experiments.

5. In Figure 3, there is no indication on the number of repetitions performed and there are no error bars for instance on Fig 3B. At least, it should be mentioned that the experiment shown is representative of x experiments. This is also true for the results shown in most figures.

Concerning the HEDS reduction assay. It is indicated in the text that this is a deglutathionylation assay and that the substrate is a glutathionylated beta-mercaptoethanol molecule. The author may refer to a paper (Begas et al, 2015 doi: 10.1039/c5sc01051a) indicating that this is probably not true and at best this is a mixture of both molecules. The way the test has been done further reduces the chance of having this molecule formed. Indeed, it is indicated that the test was started by adding HEDS and not by adding the enzyme as it should be normally done after a 3 or 5 min incubation without enzyme to allow formation of the glutathionylated beta-mercaptoethanol molecule.

For really testing deglutathionylation of such a small molecule, I suggest to incubate HEDS and GSH, way long before adding the enzyme in the assay.

Concerning the roGFP2 assay, the kinetic of the GSSG-mediated oxidation reaction catalyzed by GRX C2 appears rather slow compared to all reports on this aspect. Since it does not seem to be an effect of the concentration used (2 μ M roGFP2 vs 2.5 μ M GRX), I wonder whether this might originate from the presence of the tag. Comparing with an untagged protein would help solving this issue.

Also, it would be informative to use the ROXY CCLS variant in this assay since for many GRX removing the resolving cysteine increases their reduction/oxidation capacities.

6. Comparing the effect of GSSG on GRXC2 between fig 4 and fig 2, I do not see the same ratio of the different species, notably the fully oxidized GRXC2. It would be helpful to have an indication of the GSSG concentration or excess used in the legends.

Minor points:

- In fig. 4, indicate the masses of the molecular weight markers in all panels.

- Line 150 : there is no reference 31 in the list.

- line 211: "As mentioned above, even class II GRXs can catalyse the oxidation of roGFP2 by GSSG,..."

I see mention in the paragraph before of the incapacity to reduce roGFP2, but nothing about oxidation in the first parts of the text.

- Line 250-251 : the text indicates that the results describe a CCLS variant while the results shown in Fig. 4B are for the CSLC variant.

-Line 287-289: "We conclude that the high reactivity of class I GRXs towards GSSG is most likely due to a suitable positioning of one GSH moiety of GSSG resulting in their glutathionylation. This reaction does not seem to happen in strep-MBP-ROXY9."

I find this conclusive sentence problematic. Oxidation of strep-MBP-ROXY9 by GSSG, as seen in fig 2B, 4, necessarily implies that there is an intermediate glutathionylated form, this is just that it cannot be observed. According to that, it is written in lines 492/493 that "the glutathionylated ROXY9 is defined as unstable". Would "short-lived or transient species" be more adequate ?

-line 314 : were is repeated twice.

Line 326 high levels

Line 347 : correct : "but cannot also not perfectly be superimposed"

Line 374 : UV/Visible absorption spectrum

Line 438: should be Cys22 instead of Cys21?

Line 450 : problem with references : ref 47 refers to GRXS15, while ref 53 refers to GRXS16.

Line 464 : is almost 100% glutathionylated.

Line 681 : NEM is N-ethylmaleimide

Line 723 : correct "deglutathionylation".

In Fig S9, there should be a labelling inversion for the lanes on the left.

Nicolas Rouhier

Reviewer #2

(Remarks to the Author)

The manuscript by Mrozek et al., aims to elucidate the biochemical features of ROXY9, a GRX isoform belonging to GRX class III whose redox and catalytic properties are largely unexplored. Although past attempts to obtain the recombinant form of class III GRXs have proved unsuccessful, in this work the authors were able to obtain milligram amounts of ROXY9 from *A. thaliana* by applying the baculovirus expression system in insect cells. The recombinant protein was then used to evaluate the structural features by far-UV CD spectroscopy and to assess its ability to catalyze redox reactions (oxidase and reductase activity) and was found to be able to catalyze only the deglutathionylation of the enzyme GAPDH. Furthermore, the authors analyzed the redox state of ROXY9 following treatment with different DTTred/DTTox or GSH/GSSG ratios by alkylation with mmPEG and non-reducing SDS-PAGE analysis.

The manuscript is nicely written and clearly presented, but apart from the success in obtaining the purified protein and being able to assess its biochemical properties, it has several aspects that need to be clarified.

Major concerns:

1) Comparing the elution profile of Superdex 75 column and fraction analysis by SDS-PAGE, it is not obvious to correlate the absorption at 280 nm with the different bands attributed to the ROXY9 protein. The authors reported no data on the purification and structural analysis (i.e., gel filtration) of AtGRXC2.

2) What is the reason for the increase in secondary structures in ROXY9 and GRXC2 when analyzed in the presence of GSH? Can the authors attribute a specific secondary structure to the peaks at 210 and 224 nm? In this regard, I invite the authors to use softwares (available on the DichroWeb web server) for the estimation, even if not conclusive and precise, of the percentages of the different secondary structures based on CD spectra.

3) The formation of glutathionylated B-mercaptoethanol (B-ME-SG) requires an incubation time of 3-5 minutes (spontaneous reaction between GSH and HED, see original papers by Arne Holmgren). This was not done and the authors monitored the oxidation of NADPH just after the addition of HED.

4) The reactivation of glutathionylated GAPDH (GAPDH-SG) is quite surprising mainly because of the ROXY9:GAPDH-SG ratio reported in the manuscript. Different values are indicated and I believe that the authors should carefully revise these sections. To corroborate the hypothesis that ROXY9 can use GSH in the deglutathionylation reaction of the GAPDH enzyme, Cys mutants must be tested and compared with the wild-type form.

5) When the authors carried out redox titrations of ROXY9/GRXC2 in the presence of different GSH/GSSG ratios, they should consider the contamination of GSSG in the GSH solution when calculating redox potentials (this is clearly mentioned by the authors at page 10, lines 257-258). The GSSG contamination can be easily measured by following NADPH oxidation in the presence of GR and variable amounts of GSH. This will allow to estimate GSSG concentration and adjust redox potentials following the Nernst equation.

6) It is really surprising that the authors presented the GAPDH-SG reactivation data (see Fig. 3b and Fig. S7) by reporting histograms without the experimental data, standard deviations and statistical analyses. What is the explanation for the blocking of deglutathionylation by the addition of BPGA?

7) To prove that the structural differences revealed by comparing the AlphaFold models of ROXY9 and GRXC2 underlie the different biochemical properties, several mutants must be generated and analyzed. This is a crucial point missing from the current version of the manuscript.

8) The discussion is too long, somehow it seems a kind of mini-review. I would advise shortening it and focusing on the results presented in this work.

Minor points:

- considering the reliability of AlphaFold in determining structural models, I invite the authors to use the 3D-model to estimate solvent accessibility of Cys residues. This might be useful when comment mass spectra following exposure with reducing/oxidizing agents.
- replace GSH disulfide with glutathione disulfide
- In the abstract, it is mentioned that redox titrations were performed in the presence of different GSH/GSSG ratios while also DTT/dithiane was used leading to different results. Please comment
- In Figure 2, redox potentials of ROXY9 and GRXC2 should be read as "midpoint redox potentials"
- Why "HEDS assay" and not simply "HED assay"
- The following sentence "The release of GSSG is indirectly monitored because it is reduced by NADPH in the presence of GSH reductase (GR)" should be read "The release of GSSG is indirectly monitored because it is reduced by glutathione reductase (GR) in the presence of NADPH."

Version 1:

Reviewer comments:

Reviewer #1

(Remarks to the Author)

In their revised version of the manuscript "Molecular basis for the enzymatic inactivity 1 of class III glutaredoxin ROXY9 on standard glutathionylated substrates", the authors have adequately addressed my previous concerns either by modifying the text and/or by adding new experiments/repetitions.

There are still a few places where the text can be improved (typos or missense sentences), with a few examples below.

line 92: I see a problem with the sentence here "Figure 1b shows an alignment of selected Arabidopsis GRXs sequences important for GSH binding"

line 189 : idem "catalyse the reduction undefinded amounts, "

line 413: I guess there is a problem here : "We propose that differences in the GSH binding mode compared to that of GRXC2 are most likely responsible for the lack of enzymatic inactivity."

line 490 "It is well accepted that the interaction of GRXs with glutathione adducts in the so called scaffold site as well as the interaction of glutathionylated GRXs with GSH at the activator site are important features for oxidoreductase activity 39."

This statement comes from experiments reported in one paper. So far, evidence for the existence of an activator site remains scarce. So I would not start the sentence by it is well accepted.

line 524: "The poor GSH scaffold site of ROXY9..." What is a "poor" GSH scaffold site? Maybe think to reformulate.

Nicolas Rouhier

Reviewer #2

(Remarks to the Author)

First, I would like to thank the authors for their effort in ameliorating the manuscript based on reviewer's concerns and the current version of the work has been largely improved. However, there are still some points that need to be taken into consideration to further improve the manuscript:

1) In the abstract, the sentence referring to the redox response of AtGRXC2/HsGRX1 to different GSH/GSSG ratios is somewhat confusing when compared with that of ROXY9. Specifically, what is the meaning "over a wide range of GSH/GSSG ratios"? This should refer to redox potentials and not simply ratios of the glutathione molecules. In the reviewer's opinion, it is (perhaps) clear the authors' intention to emphasize the difference between Class III GRXs and Class I GRXs, but this may not be clear at this stage of the reading.

2) In Figure 1A, please label Tyr as done for Cys residues and also indicate N- and C-ter of the AF model.

3) Did the authors carry out gel filtration analysis on the C-to-S variants? This should be integrated in the main text of the current version of the manuscript as it might be helpful in elucidating the molecular mechanisms underlying the formation of intermolecular disulfide(s) responsible for dimerization/trimerization of the protein (see lines 131-136). I suggest simply focusing on the Cys variants already in the manuscript.

4) The pH value should be indicated when referring to redox couples (e.g., lines 156-157).

5) The number of Cys residues should be mentioned and potentially before mmPEG alkylation experiments. Despite mentioned in the Methods section, the authors should indicate that mmPEG-dependent alkylation occurs under denaturing conditions thus exposing to the label all Cys residues. It is indeed possible to label Cys residues under native conditions and then quench mmPEG before protein denaturation thus labelling only solvent accessible Cys.

6) The HED assay is not the one described by the authors. I suggest reading original papers and all the others that employed this technique to evaluate the deglutathionylation activity of GRX. Lines 188-191 must be revised accordingly.

7) I can understand the authors' efforts to optimize the glutathionylated GAPDH reduction assay. In this respect, the author should consider performing this assay using GRX in the presence of GSH (2 mM is fine) in the presence of NADPH and GR. This will be extremely important to completely reduce GRX and allow for improved reduction efficiency by avoiding the interference of GSSG present in GSH and its effect on the redox state of GRX. This would certainly apply to GRXC2 and will be instrumental to verify that class I GRXs are more efficient than class III to catalyze deglutathionylation reactions. Based on Figure 3d, this is not observed and not coherent with previous studies on cytoplasmic GAPDH from Arabidopsis thaliana (AtGAPC1).

8) The enzymatic assay for GAPDH is extremely complicated, and the authors should consider that BPGA is an unstable compound. Why not use a BPGA-generating system (PGK plus ATP and 3PGA) in the cuvette also containing GAPDH and NADH? This system is normally used and, in my personal experience, is extremely efficient. Another tip concerns enzyme measurement. It is not optimal to proceed to obtain deltaAbs data only every minute. For this I recommend using a classical UV/Vis spectrophotometer and not a plate reader as I think was used.

9) For the purpose of further specifying the redox mechanism of GRX3 in *Chlamydomonas reinhardtii*, the authors should consider that during the process of deglutathionylation of photosynthetic GAPDH (GAPA), an intramolecular disulfide bridge is formed after the transfer of glutathione moiety from the protein target to GRX3, which is subsequently reduced by the FDX/TRX system.

10) Line 715. Remove GSSG as it was not used

We thank both reviewers for carefully reading the manuscript and for their suggestions to improve it. Please find below our comments how the concerns were addressed (marked in yellow). Line numbers refer to the re-submitted manuscript without track changes. Phrases that can be found in the novel manuscript are in red.

REVIEWER COMMENTS

Reviewer #1 (Remarks to the Author):

The paper entitled “Molecular basis for the enzymatic inactivity of class III glutaredoxins on standard glutathionylated substrates” by Mrozek and collaborators focuses on the biochemical and structural properties of a representative member, ROXY9, of the expanded class III glutaredoxins (GRX) from plants. These GRX usually possess a CCxC signature that differs from the classical CPYC signature present in class I GRX. A comparative analysis of the *in vitro* redox properties and activity profile of ROXY9 was provided using the class I GRXC2 from *Arabidopsis* as a reference.

It is shown that, unlike GRXC2, ROXY9 is inactive as a reductase on the small substrate HEDS and as an oxidase on roGFP2. However, both proteins possess the capacity to catalyze the deglutathionylation of GAPDH, although to different extent. Redox titrations revealed the formation of a disulfide bond between the first and the last cysteine of the CCLC signature with a value comprised between -220 to -230 mV, as found in many other GRX. Moreover, the determination of the pKa of the catalytic cysteine indicates that it is not different either from other characterized GRX. The GSSG-mediated glutathionylation of the three other cysteine residues present in ROXY9 is also evidenced but not analyzed in detail. Reduction-Oxidation experiments using DTT (oxidized or reduced), GSH (oxidized, reduced or in mixtures of defined redox potentials) point to differences between ROXY9 and AtGRXC2. It is observed that GRXC2 is oxidized to roughly equal amounts of a glutathionylated form (GRX-SG) and a form with an intramolecular disulfide (GRX-S2) over a wide range of GSH/GSSG ratios whereas ROXY9 readily forms an intramolecular disulfide and glutathionylation of the first reactive cysteine was not observed, suggesting that its formation is transient. From structural modeling, it is hypothesized that this different reactivity would be due to subtle differences in the GSH binding mode and that this feature might have evolved to avoid overlapping functions with class I GRXs in planta.

Overall, this study focuses on an important and highly relevant question that deserves to be explored since the analysis of the oxidoreductase activity of these specific GRX was hampered in most cases in the past by the inability to express recombinant proteins. There are substantial amounts of experiments supporting the conclusions. While most experiments have been well achieved and the results discussed with respect to the known literature, I have quite a number of concerns, comments and suggestions for some of them.

Introduction

- lines 70-72: “When functioning as a reductase of glutathionylated substrates, the GSH moiety of the substrate has to be positioned into the GRX GSH binding groove so that the thiol points towards

CysA.”

In this context, glutathione forms a disulfide with a protein cysteine residue. So, I would avoid (i) using GSH (here and at any places where it does not refer to reduced glutathione) which is used for the reduced form of glutathione and (ii) saying that the thiol points towards CysA since it is not in the thiol form.

We corrected this:

69ff: When functioning as a reductase of glutathionylated substrates, the glutathione moiety of the substrate has to be positioned into the GRX GSH binding groove so that the sulphur points towards the thiol group of Cys_A...

-Line 94: remove the f after the references 13 and 14.

Corrected

- Lines 94-96 : “Class III GRXs do not encode the class II-specific five amino acid loop which interferes with oxidoreductase activity nor the proline in the active site which might interfere with FeS cluster assembly.”

Please provide references again for the first aspect.

We cited Liedgens et al. (2020), and Trnka et al (2020) for the analysis of the loop structure, line 94

For the question of the proline, I assume that this refers to class I GRX and notably the work performed with GRX from poplar (Rouhier et al 2007, doi: 10.1073/pnas.0702268104).

We cited Rouhier et al., 2007 in line 94

Results

1. It is interesting that a class III GRX can be produced in an insect cell system, even though it is fused to a strep-MBP tag.

I have several comments here. It would be informative to know whether the protein is expressed with an Fe-S cluster, and whether it is purified in a reduced or oxidized form.

As mentioned in the MM section, we purified the proteins in the presence of 5 mM GSH. As judged from the colour, there is no FeS cluster. We performed several FeS reconstitution experiments with Roland Lill in Marburg under anaerobic conditions, where we found dimerization of the protein but the spectra and Fe and S measurements were not at all convincing with respect to identification of a cluster. We invested a lot of work, but data were not good enough to integrate them into the manuscript. We changed the text as indicated below.

125ff: After affinity purification **in the presence of 5 mM GSH** (Supplementary Fig. 1a), 25 to 40 mg of both proteins were routinely obtained from 600 ml of Hi5 cells. **Despite the high protein concentration (100 to 200 µM), samples lacked the brownish colour indicative for FeS clusters.**

Indeed, it is mentioned that a gel filtration performed with DTT indicates a monomeric state, but the reader would be tempted to conclude that DTT may be present to avoid oligomerization.

We now present the outcome of gel filtration experiments after we had changed our purification protocol using GSH instead of DTT. This is shown in Supplementary Fig. S1. We changed the text accordingly.

128 ff Analytical gel filtration analysis documented that at least half of the proteins eluted as aggregates/higher oligomers of > 670 kDa (Supplementary Fig. 1b-d). The non-aggregated protein species appeared to elute as monomers, dimers and trimers in case of strep-MBP-ROXY9 (Supplementary Fig. 1b) and as monomers in case of strep-MBP-GRXC2 (Supplementary Fig. 1c). DTT treatment of strep-MBP-ROXY9 led to a shift from oligomers to mostly monomers (Supplemental Fig. 1d), indicating that the protein is prone to forming intermolecular disulfides even in the presence of 5 mM GSH. The very high molecular weight aggregates could not be resolved by treatment with DTT.

Also, it is shown that it is possible to remove the tag and keep a soluble protein. But there is no mention of the oligomeric state and stability of the protein. In fact, it has to be justified why such a bulky tagged protein is used throughout the study, because using an untagged protein would be much more relevant.

Answer: Preparation of the untagged GRXs was done by TEV cleavage of the strep-MBP-GRX fusion proteins and subsequent gel filtration. Especially for ROXY9, yields were low (incomplete cleavage, low yield in the following purification steps). Still we managed to perform the CD spectra and one HED assay (Supplemental Fig. S5). The redox assay based on Coomassie Brilliant Blue staining would require a lot of protein because of the low molecular weight of GRXs. Since we were also concerned about the bulky strep-MBP tag, we analysed His-tagged proteins at least for the critical redox titration experiment. We have included the redox titrations in the new Supplementary Fig. 17, which yield a similar picture: In freshly prepared GSH, GRXC2 is readily glutathionylated and forms an intramolecular disulfide, while ROXY9 remained primarily in the reduced state.

2. The results of the CD analysis (Figure c) puzzle me, since it is indicated that adding GSH increases the secondary structure content. It is rather unexpected if we refer to known GRX structures that were solved with or without bound glutathione and for which there are no change visible. Could it be that the effect is due to the strepMBP tag? In principle, the changes in the secondary structure content can be quantified.

First of all, the CD spectra were made with the unfused proteins. To be honest, I found the results also very puzzling. Finally, we found out that the amounts of protein that were analysed had been miscalculated. Now, the data were normalized utilizing the absorbance at 220 nm obtained in parallel to circular dichroism data. We changed the text in the results section and in the methods part.

140ff The CD spectra of both proteins were almost identical, independent of whether measurements were made in the presence of DTT or GSH (Fig. 1c)

647ff: „Due to the poor extinction coefficient at 280 nm, determination of protein concentration at the concentrations employed proved inaccurate. Instead, we normalized the data utilizing the absorbance at 220 nm obtained in parallel to circular dichroism data.”

3. Concerning the experiments about the thermal stability, it would be informative to examine the redox state of the proteins before analysis and compare reduced and oxidized forms and even as-purified proteins.

Thermal stability was assessed in the presence of 1 mM DTT, presumably resulting in the reduced forms of the proteins. We added this information in the legend of Supplementary Fig. 2b.

We admit that it would be interesting to see that oxidation leading to disulfide bond formation would increase the stability. We wonder, whether this has been done for other GRXs. However, the aim of the experiment was to indirect evidence for the assumption that ROXY9 adopts a stable thioredoxin fold.

142 ff Both proteins denatured at a melting temperature of 70.6°C and 69.2°C, respectively (Supplementary Fig. 2a), which is consistent with the previously reported high thermal stability of other GRXs (13, 30).

4. Alkylation experiments using 5kDa mmPEG have been used extensively here to evaluate the redox properties of the proteins. However, the retarded migration observed on SDS PAGE gel is much larger than expected (differences of approx. 70 kDa instead of 25 kDa). This has to be at least discussed. Are there cysteine residues in the tag?

Answer: This phenomenon was described before by Couturier et al. (2013); we mention this now in the manuscript. We also mentioned that MBP does not contain any cysteines.

158ff Upon treatment of strep-MBP-ROXY9 with 10 mM DTT and subsequent alkylation, the mobility of the protein was reduced due to the addition of mmPEG to the five reduced cysteines in the ROXY9 moiety of the protein (Fig. 2). The shift observed following the alkylation was larger than expected, a phenomenon that has been described before and might be due to the interaction of the mmPEG with the polyacrylamide matrix (ref 31).

Evaluation of the effect of dithiane on the proteins, demonstrating the oxidation of only the two cysteine residues forming an intramolecular disulfide on ROXY9 or GRXC2, has been shown only in fig. 5, while I would have expected to see the results prior other experiments.

This is correct. Still, the analysis of the SCLC, CSLC and CCLS variants (Supplementary Figure 4) support the conclusion that the disulfide is formed between CysA and CysB within the active site motif as shown in many CPYC-type GRXs before. Please note that the second cysteine in CC type GRXs renders the mass spec data not entirely conclusive.

5. In Figure 3, there is no indication on the number of repetitions performed and there are no error bars for instance on Fig 3B. At least, it should be mentioned that the experiment shown is representative of x experiments. This is also true for the results shown in most figures.

We now invested a lot of work in showing error bars for the enzymatic assays. All the assays shown in Fig. 3 are now from 3 independent biological replicates (i.e. three different batches of insect cells yielding three independent protein preparations). We describe the origin of the error bars in the legend. In our hands, the GAPDH assay turned out to be hard to establish, but it confirmed that ROXY9 has a GSH-dependent de-glutathionylation activity, which is clearly lower than the corresponding GRXC2 activity. We also reproduced the most critical redox titration experiment with His-tagged proteins, yielding the same result as observed with Strep-MBP-derivatives (Supplementary Fig. 17).

Concerning the HEDS reduction assay. It is indicated in the text that this is a deglutathionylation assay and that the substrate is a glutathionylated beta-mercaptoethanol molecule. The author may refer to a paper (Begas et al, 2015 doi: 10.1039/c5sc01051a) indicating that this is probably not true and at best this is a mixture of both molecules. The way the test has been done further reduce the chance of having this molecule formed. Indeed, it is indicated that the test was started by adding

HEDS and not by adding the enzyme as it should be normally done after a 3 or 5 min incubation without enzyme to allow formation of the glutathionylated beta-mercaptoethanol molecule. For really testing deglutathionylation of such a small molecule, I suggest to incubate HEDS and GSH, way long before adding the enzyme in the assay.

We now followed a standard protocol and pre-incubated HEDS and GSH for 5 min before adding the protein. Moreover, we included an assay using glutathionylated cysteine, which starts with a clearly defined amount of glutathionylated substrate.

We tried to be more accurate when describing the HED assay.

187ff In this assay, GRXs first catalyse the reduction of undefined amounts of the disulfide in HED with GSH as electron donor, leading to β -mercaptoethanol (β -ME) and glutathionylated β -ME (β -ME-SG), the latter one serving as a substrate for the de-glutathionylation reaction (Begas et al., 2015).

Concerning the roGFP2 assay, the kinetic of the GSSG-mediated oxidation reaction catalyzed by GRX C2 appears rather slow compared to all reports on this aspect. Since it does not seem to be an effect of the concentration used (2 μ M roGFP2 vs 2.5 μ M GRX), I wonder whether this might originate from the presence of the tag. Comparing with an untagged protein would help solving this issue. Also, it would be informative to use the ROXY CCLS variant in this assay since for many GRX removing the resolving cysteine increases their reduction/oxidation capacities.

We were also wondering about the slow kinetics. When changing the buffer to 100 mM Tris/HCl pH 8, the kinetics became faster (see Fig. 3). We also noticed that strep MBP-GRX quenches the fluorescence. We included 2 μ M strep-MBP-GRX in the reference sample with oxidized roGFP. We also analysed the CCLS variant in all our assays.

196 ff: Since several class I GRXs become more active when mutated for their resolving cysteine residues (references 40-42) we changed the ROXY9 CCLC signature into CCLS.

6. Comparing the effect of GSSG on GRXC2 between fig 4 and fig 2, I do not see the same ratio of the different species, notably the fully oxidized GRXC2.

Indeed, the degree of glutathionylation of the cysteine outside the active center might vary. In the current version, we added another Supplemental Fig. 17 with His-tagged GRXC2, which shows a pattern similar to Fig. 2.

It would be helpful to have an indication of the GSSG concentration or excess used in the legends.

We changed the legends accordingly.

Minor points:

- In fig. 4, indicate the masses of the molecular weight markers in all panels.

Corrected

- Line 150 : there is no reference 31 in the list.

We corrected this. The mistake must have occurred when editing the list of references as provided by endnote.

- line 211: "As mentioned above, even class II GRXs can catalyse the oxidation of roGFP2 by GSSG,..."

I see mention in the paragraph before of the incapacity to reduce roGFP2, but nothing about oxidation in the first parts of the text.

We rewrote this paragraph. We also show an experiment where we show that ROXY9 fails – in contrast to GRXC2 – to reduce oxidized roGFP2. (Supplemental Fig. 7).

223ff Even class II GRXs AtGRXS15, AtGRXS16 and HsGRX5 can catalyse the oxidation of roGFP2 by GSSG, even though they fail to mediate its reduction

- Line 250-251 : the text indicates that the results describe a CCLS variant while the results shown in Fig. 4B are for the CSLC variant.

We corrected this. It should have been 4c.

-Line 287-289: “We conclude that the high reactivity of class I GRXs towards GSSG is most likely due to a suitable positioning of one GSH moiety of GSSG resulting in their glutathionylation. This reaction does not seem to happen in strep-MBP-ROXY9.”

I find this conclusive sentence problematic. Oxidation of strep-MBP-ROXY9 by GSSG, as seen in fig 2B, 4, necessarily implies that there is an intermediate glutathionylated form, this is just that it cannot be observed. According to that, it is written in lines 492/493 that “the glutathionylated ROXY9 is defined as instable”. Would “short-lived or transient species” be more adequate ?

We deleted this conclusion from the result parts, and wrote in the discussion:

503 ff: In ROXY9, the glutathionylated protein species is very short-lived, which explains the lack of oxidase activity of ROXY9 on roGFP2.

-line 314 : were is repeated twice.

Corrected

Line 326 high levels

corrected

Line 347 : correct : “but cannot also not perfectly be superimposed”

corrected

Line 374 : UV/Visible absorption spectrum

Corrected

Line 438: should be Cys22 instead of Cys21?

Corrected

Line 450 : problem with references : ref 47 refers to GRXS15, while ref 53 refers to GRXS16.

The references belong to our previous discussion, which mentions speculations in the literature that class II GRXs might not function as reductases because they might not become reduced by GSH, i.e. that their activator site might not be optimal. Since our work rather focussed on the comparison of class I and class III glutaredoxins, and since data on this issue vary depending on the protein we decided to delete this part of our discussion.

Line 464 : is almost 100% glutathionylated.

Corrected

Line 681 : NEM is N-ethylmaleimide

Corrected

Line 723 : correct "deglutathionylation".

Since we carefully revisited the GAPDH assay, we had to rephrase this section anyways.

In Fig S9, there should be a labelling inversion for the lanes on the left.

Corrected

Nicolas Rouhier

Reviewer #2 (Remarks to the Author):

The manuscript by Mrozek et al., aims to elucidate the biochemical features of ROXY9, a GRX isoform belonging to GRX class III whose redox and catalytic properties are largely unexplored. Although past attempts to obtain the recombinant form of class III GRXs have proved unsuccessful, in this work the authors were able to obtain milligram amounts of ROXY9 from *A. thaliana* by applying the baculovirus expression system in insect cells. The recombinant protein was then used to evaluate the structural features by far-UV CD spectroscopy and to assess its ability to catalyze redox reactions (oxidase and reductase activity) and was found to be able to catalyze only the deglutathionylation of the enzyme GAPDH. Furthermore, the authors analyzed the redox state of ROXY9 following treatment with different DTTred/DTTox or GSH/GSSG ratios by alkylation with mmPEG and non-reducing SDS-PAGE analysis.

The manuscript is nicely written and clearly presented, but apart from the success in obtaining the purified protein and being able to assess its biochemical properties, it has several aspects that need to be clarified.

Major concerns:

1) Comparing the elution profile of Superdex 75 column and fraction analysis by SDS-PAGE, it is not

obvious to correlate the absorption at 280 nm with the different bands attributed to the ROXY9 protein.

Since ROXY9 does not contain a Trp, tracking of the protein by UV spectroscopy does not give high signals. We found increased absorption in 7 fractions and loaded aliquots from those on a gel. This yielded a band with the expected molecular weight of 12 kDa.

The authors reported no data on the purification and structural analysis (i.e., gel filtration) of AtGRXC2.

128 ff Analytical gel filtration analysis documented that at least half of the proteins eluted as aggregates/higher oligomers of > 670 kDa (Supplementary Fig. 1b-d). The non-aggregated protein species appeared to elute as monomers, dimers and trimers in case of strep-MBP-ROXY9 (Supplementary Fig. 1b) and as monomers in case of strep-MBP-GRXC2 (Supplementary Fig. 1c). DTT treatment of strep-MBP-ROXY9 led to a shift from oligomers to mostly monomers (Supplemental Fig. 1d), indicating that the protein is prone to forming intermolecular disulfides even in the presence of 5 mM GSH. The very high molecular weight aggregates could not be resolved by treatment with DTT.

2) What is the reason for the increase in secondary structures in ROXY9 and GRXC2 when analyzed in the presence of GSH? Can the authors attribute a specific secondary structure to the peaks at 210 and 224 nm? In this regard, I invite the authors to use softwares (available on the DichroWeb web server) for the estimation, even if not conclusive and precise, of the percentages of the different secondary structures based on CD spectra.

This point was also raised by reviewer #1 and we have to admit an embarrassing mistake: the amounts of protein that had been used for the assay had been miscalculated. Now, the data were normalized utilizing the absorbance at 220 nm obtained in parallel to circular dichroism data. We changed the text in the results section and in the methods part.

140ff The CD spectra of both proteins were almost identical, independent of whether measurements were made in the presence of DTT or GSH (Fig. 1c)

647ff: „Due to the poor extinction coefficient at 280 nm, determination of protein concentration at the concentrations employed proved inaccurate. Instead, we normalized the data utilizing the absorbance at 220 nm obtained in parallel to circular dichroism data.”

3) The formation of glutathionylated B-mercaptoethanol (B-ME-SG) requires an incubation time of 3-5 minutes (spontaneous reaction between GSH and HED, see original papers by Arne Holmgren). This was not done and the authors monitored the oxidation of NADPH just after the addition of HED.

This point was also raised by reviewer #1. We now followed a standard protocol and pre-incubated HEDS and GSH for 5 min before adding the protein. Moreover, we included the GSSCys assay (Fig. 3b), which starts with clearly defined amounts of glutathionylated substrate.

4) The reactivation of glutathionylated GAPDH (GAPDH-SG) is quite surprising mainly because of the ROXY9:GAPDH-SG ratio reported in the manuscript. Different values are indicated and I believe that the authors should carefully revise these sections. To corroborate the hypothesis that ROXY9 can use GSH in the deglutathionylation reaction of the GAPDH enzyme, Cys mutants must be tested and compared with the wild-type form.

We revised this section. We invested considerable work in the assay to get more reproducible results with error bars resulting from three independent protein preparations. We confirmed our previous results showing deglutathionylation activity only in the presence of GSH. Note, that in this assay, only 6% of the GAPDH fraction which can be reduced with DTT is deglutathionylated in 30 min although there was a 5-fold molar excess of ROXY9 over the GAPDH in the assay. Assuming that the glutathione moiety of GAPDH is transferred to the reactive cysteine of ROXY9, there is no need to regenerate the enzyme with GSH. Still, we require GSH in the assay. In case the reaction goes via a transiently glutathionylated ROXY9, we would have to postulate that ROXY9 can only obtain the glutathione if a GSH is bound to e.g. the “activator” site. Alternatively and maybe more likely, ROXY9 “activates” a GSH for the disulfide exchange reaction as recently proposed for the PfGRX-catalysed reduction of roGFP2(S2) to roGFP-SG. Geissel et al. 2024.(Fig. 7b).

Even GRXC2 activity is weak, we are very close to background. The properties of the ROXY9 did not change when we replaced the CCLC motif by CPYC or CCLS, suggesting that the first cysteine is responsible for the catalysis or “activation” of GSH, respectively. In all other assays, which stringently require a glutathionylated intermediate, ROXY9 is inactive.

We discuss the results in the current version:

511ff Still, we discovered weak GSH-dependent reductase activity with GAPDH-SG. In our assay, 2.5 μM strep-MBP-ROXY9 deglutathionylated 0.018 μM GAPDH in 20 min. At this ratio of enzyme over substrate, the reaction should not require regeneration of the enzyme by GSH. The requirement for GSH can be better explained by assuming that ROXY9 weakly binds GSH in the scaffold site, leading to its deprotonation and thus to its activation (45, 60). This activated GS- might attack the glutathione mixed disulfide of GAPDH-SG, leading to GAPDH and GSSG. Likewise, GRXC2, which might cycle between GRXC2-SG and GRXC2-S2 in the absence of GR and NADPH, might provide activated GS-. In contrast, the deglutathionylation activity mediated by class II C. reinhardtii GRX3 was independent from GSH. In this case, GAPDH recovery can be explained by the transfer of glutathione to GRX3, which was maintained in the reduced state by light-driven electron transport through photosystem I 49.

5) When the authors carried out redox titrations of ROXY9/GRXC2 in the presence of different GSH/GSSG ratios, they should consider the contamination of GSSG in the GSH solution when calculating redox potentials (this is clearly mentioned by the authors at page 10, lines 257-258). The GSSG contamination can be easily measured by following NADPH oxidation in the presence of GR and variable amounts of GSH. This will allow to estimate GSSG concentration and adjust redox potentials following the Nernst equation.

We determined the amount of GSSG in freshly prepared GSH as suggested. For comparison, we used 5, 10, 20 and 30 μM GSSG. We determined the concentration was 24 μM GSSG in a 11 mM GSH sample. Assuming even 30 μM GSSG i.e. 0.27% in freshly prepared GSH, we obtain the following values for the redox potentials we used.

Instead of -250: -239

Instead of -230: -227

Instead of -220: -218

Instead of -210: -209

Instead of -200: -199

Freshly prepared GSH would have a redox potential of -247.

The values for the midpoint redox potentials are not affected by the GSSG contamination.

Since the percentage might vary in the experiments, depending of how fresh the GSH was (it was always prepared on the same day, but still.....), we would like to refrain from precise values. We admit that our data are more qualitative than quantitative. The main point we would like to make is that the GSSG/GSH midpoint redox potential of ROXY9 is similar to the DTT/dithiane midpoint redox potential, while this is dramatically different for the two class I GRXs, which get oxidized by lower GSSG/GSH midpoint redox potentials and show a similar ratio of glutathionylated and disulfide-containing species over a wide range of redox potentials. We have added a novel Supplemental Fig. 16 with His-tagged proteins to support these data.

6) It is really surprising that the authors presented the GAPDH-SG reactivation data (see Fig. 3b and Fig. S7) by reporting histograms without the experimental data, standard deviations and statistical analyses. What is the explanation for the blocking of deglutathionylation by the addition of BPGA?

For quality and reproducibility of the assay, please see our comments above (point 4).

Concerning the blocking of deglutathionylation by the addition of BPGA, we made a mistake. In our assay it just serves as a substrate.

7) To prove that the structural differences revealed by comparing the AlphaFold models of ROXY9 and GRXC2 underlie the different biochemical properties, several mutants must be generated and analyzed. This is a crucial point missing from the current version of the manuscript.

I can completely understand this suggestion and I was tempted to start a project with chimeric proteins and our analysis together with the knowledge on class I and class II GRX has paved the ground. It seems obvious to start with Glu/Gln exchanges in helix 2 and the replacement of the whole loop by a class I loop. However, other amino acids in class I GRXS might contribute to the relative position of the structural elements of the thioredoxin fold so that it binds glutathionylated substrates in a productive way. Eorking with recombinant proteins produced in insect cells is quite time consuming and distortions of the protein fold might lead to low yields. I have heard that Bruce Morgan used roGFP2-GRX fusion constructs for the assessment of Grx structure–function relationships inside living yeast cells. he had a much faster read-out. I do not know the scientific details, but he tried to change a class III GRX into a class I GRX by interconversion mutants and failed. When the groups of Deponte and Lillig tried to change enzymatically inactive class II GRXs into class I GRXs by altering the most obvious structural elements (5 aa loop and active site) only a few percent of enzymatic activity could be recovered. It is definitely worth a try, but it might take some time and is unfortunately beyond the scope of this manuscript.

8) The discussion is too long, somehow it seems a kind of mini-review. I would advise shortening it and focusing on the results presented in this work.

We managed to reduce the word count of the discussion from 3686 to 2948 words

Minor points:

- considering the reliability of AlphaFold in determining structural models, I invite the authors to use the 3D-model to estimate solvent accessibility of Cys residues. This might be useful when comment mass spectra following exposure with reducing/oxidizing agents.

We added Supplementary Table 1 for the data on the solvent accessibility of Cys residues. The values were not really helpful for explaining the degree of the oxidation of cysteine residues outside the active site, but we were able to integrate this information twice in the manuscript.

151 ff Typically, the catalytic cysteine is exposed to the solvent, while the resolving cysteine is buried, a pattern that is also observed for GRXC2 and ROXY9 (Supplementary Table 1).

266 ff Thus, glutathionylation of Cys22, which has a large exposed surface area (Supplementary Table 1), seems to occur in strep-MBP-ROXY9-S2.

- replace GSH disulfide with glutathione disulfide

Corrected

- In the abstract, it is mentioned that redox titrations were performed in the presence of different GSH/GSSG ratios while also DTT/dithiane was used leading to different results. Please comment

We did not mention the DTT/dithiane redox titrations due to word count limitations.

We

- In Figure 2, redox potentials of ROXY9 and GRXC2 should be read as “midpoint redox potentials”

Corrected

- Why “HEDS assay” and not simply “HED assay”

Both is found in the literature, but we changed to HED.

- The following sentence “The release of GSSG is indirectly monitored because it is reduced by NADPH in the presence of GSH reductase (GR)” should be read “The release of GSSG is indirectly monitored because it is reduced by glutathione reductase (GR) in the presence of NADPH.

This part was rephrased...

190 GSSG is subsequently reduced by glutathione reductase (GR) in the presence of NADPH

We thank both reviewers for again taking the time to carefully read the manuscript and for pointing out deficiencies. Please find below our comments how the concerns were addressed (marked in yellow). Line numbers refer to the re-submitted manuscript. Phrases that can be found in the novel manuscript are in red.

In the manuscript with track changes, we marked changes requested by the reviewers in red, changes that appeared to be helpful after re-reading the manuscript are marked in blue

REVIEWER COMMENTS

Reviewer #1 (Remarks to the Author):

In their revised version of the manuscript "Molecular basis for the enzymatic inactivity 1 of class III glutaredoxin ROXY9 on standard glutathionylated substrates", the authors have adequately addressed my previous concerns either by modifying the text and/or by adding new experiments/repetitions.

There are still a few places where the text can be improved (typos or missense sentences), with a few examples below.

line 92:I see a problem with the sentence here "Figure 1b shows an alignment of selected Arabidopsis GRXs sequences important for GSH binding"

We replaced this sentence by:

93ff: Previous structural studies of class I and class II GRXs from different organisms had identified several amino acid residues that are involved in glutathione binding. The amino acid environments of these residues as found in sequences representing all three GRX classes encoded in the Arabidopsis genome are shown in Fig. 1b.

line 189 : idem "catalyse the reduction undefinded amounts, "

We were not sure, whether the reviewer pointed out the typing error (of undefined amounts rather than undefinded amounts). We were not happy with this expression anyways. Consequently, it was deleted.

line 413: I guess there is a problem here : "We propose that differences in the GSH binding mode compared to that of GRXC2 are most likely responsible for the lack of enzymatic inactivity."

Indeed, it should be lack of enzymatic activity. We corrected this.

line 490 "It is well accepted that the interaction of GRXs with glutathione adducts in the so called scaffold site as well as the interaction of glutathionylated GRXs with GSH at the activator site are important features for oxidoreductase activity 39."

This statement comes from experiments reported in one paper. So far, evidence for the existence of an activator site remains scarce. So I would not start the sentence by it is well accepted.

We deleted the aspect of the activator site, because it does not play a role in our explanation of the lacking oxidoreductase activity of ROXY9.

line 524: "The poor GSH scaffold site of ROXY9..." What is a "poor" GSH scaffold site? Maybe think to reformulate.

Since subheadings in the Discussion section are not allowed in Nature Comm, we had to delete this sentence.

Reviewer #2 (Remarks to the Author):

First, I would like to thank the authors for their effort in ameliorating the manuscript based on reviewer's concerns and the current version of the work has been largely improved. However, there are still some points that need to be taken into consideration to further improve the manuscript:

1) In the abstract, the sentence referring to the redox response of AtGRXC2/HsGRX1 to different GSH/GSSG ratios is somewhat confusing when compared with that of ROXY9. Specifically, what is the meaning "over a wide range of GSH/GSSG ratios"? This should refer to redox potentials and not simply ratios of the glutathione molecules. In the reviewer's opinion, it is (perhaps) clear the authors' intention to emphasize the difference between Class III GRXs and Class I GRXs, but this may not be clear at this stage of the reading.

Given that we had to reduce the number of words to 150, we rephrased the abstract at the expense of giving detailed experimental results.

Class I glutaredoxins (GRXs) are nearly ubiquitous proteins that catalyse the glutathione (GSH)-dependent reduction of mainly glutathionylated substrates. In land plants, a third class of GRXs has evolved (class III). Class III GRXs regulate the activity of TGA transcription factors through yet unexplored mechanisms. Here we show that *Arabidopsis thaliana* class III GRX ROXY9 is inactive as an oxidoreductase on widely used model substrates. Glutathionylation of the active site cysteine, a prerequisite for enzymatic activity, occurs only under highly oxidizing conditions established by the GSH/glutathione disulfide (GSSG) redox couple, while class I GRXs are readily glutathionylated even at very negative GSH/GSSG redox potentials. Thus, structural alterations in the GSH binding site leading to an altered GSH binding mode likely explain the enzymatic inactivity of ROXY9. This might have evolved to avoid overlapping functions with class I GRXs and raises questions of whether ROXY9 regulates TGA substrates through redox regulation.

2) In Figure 1A, please label Tyr as done for Cys residues and also indicate N- and C-ter of the AF model.

We corrected this.

3) Did the authors carry out gel filtration analysis on the C-to-S variants? This should be integrated in the main text of the current version of the manuscript as it might be helpful in elucidating the molecular mechanisms underlying the formation of intermolecular disulfide(s) responsible for dimerization/trimerization of the protein (see lines 131-136). I suggest simply focusing on the Cys variants already in the manuscript.

The tendency of ROXY9 to form intermolecular disulfides can also be observed at the top of the non-reducing SDS PAGES (Fig. 1, Fig. 4). These oligomers are observed for all the active site mutants but not for the CSSSS mutant (Fig. 6), suggesting that intermolecular disulfide formation is not due to the

class III-specific active site, but rather the two cysteines outside the active site. We mentioned this now in the current manuscript.

170ff: Moreover, the amount of protein species with very low electrophoretic mobility increased, again unravelling the tendency of the protein to form intermolecular disulfides as also already revealed by size exclusion chromatography (Supplementary Fig. 1).

357ff: Of note, the strep-MBP-ROXY9-CSSSS variant did not form the disulfide-bridged oligomers, which were observed with the CCLC (wt), SCLC, CSLC, CCLS variants (Fig. 1 and Supplemental Fig. 4), suggesting that the cysteines outside the active site contribute to oligomerisation.*

***Sorry for retracting the last sentence after uncropping of the gel as supplied in the Source Data.**

4) The pH value should be indicated when referring to redox couples (e.g., lines 156-157).

Corrected

Lines 122, 160, 175, 177, 189, 263, 423, 425, 462, 463, 470, 471

5) The number of Cys residues should be mentioned and potentially before mmPEG alkylation experiments. Despite mentioned in the Methods section, the authors should indicate that mmPEG-dependent alkylation occurs under denaturing conditions thus exposing to the label all Cys residues. It is indeed possible to label Cys residues under native conditions and then quench mmPEG before protein denaturation thus labelling only solvent accessible Cys.

Somehow I could not follow the first suggestions without eliciting major rearrangements in the text, but I imagine that the information is still placed in an appropriate position.

We mentioned the conditions of alkylation in the main text. Alkylation under denaturing conditions was done on purpose in order to deal with one parameter only, i.e. redox state independent of solvent accessibility.

165/166: ...subsequent alkylation of the TCA-precipitated protein in the presence of 1% SDS, the mobility...

6) The HED assay is not the one described by the authors. I suggest reading original papers and all the others that employed this technique to evaluate the deglutathionylation activity of GRX. Lines 188-191 must be revised accordingly.

Indeed, in former reference 38, HED is used as an inhibitor and not as a substrate. Therefore, I agree, that the reference is not appropriate, although it shows that GRX can form a mixed disulfide with HED. I deleted the reference here. I also deleted former reference 40, because it did not describe the HED assay. Instead, I chose to cite the following manuscripts:

(39) Nagai et al., 1962 (according to Begas et al., 2015 one of the first descriptions, reaction was started with GSH)

(40) Mieyal et al., 1991, reaction was started with HEDS

(41) Zaffagnini et al., 2008, reaction was started with GRX

(42) Begas et al., 2015

7) I can understand the authors' efforts to optimize the glutathionylated GAPDH reduction assay. In

this respect, the author should consider performing this assay using GRX in the presence of GSH (2 mM is fine) in the presence of NADPH and GR. This will be extremely important to completely reduce GRX and allow for improved reduction efficiency by avoiding the interference of GSSG present in GSH and its effect on the redox state of GRX. This would certainly apply to GRXC2 and will be instrumental to verify that class I GRXs are more efficient than class III to catalyze de-glutathionylation reactions. Based on Figure 3d, this is not observed and not coherent with previous studies on cytoplasmic GAPDH from *Arabidopsis thaliana* (AtGAPC1).

In Figure 3d, we had to use about 10-fold more ROXY9 than GRXC2 to obtain similar activities, but this difference would be probably bigger if we would have performed the assay in the presence of GR and NADPH. To our excuse, I would like to mention that we followed the protocol of Zaffagnini et al., 2008, where inclusion of GR and NADPH was not mentioned. I assumed that the reason for omission of these compounds was due to a potential contribution of NADPH and GR to the decrease of absorbance at 340 nm even after the addition of the GAPDH substrate, since de-glutathionylation would proceed. However, when reading Bedhomme et al., I realized that there seems to be no interference. I consider it still remarkable that the assay works under our conditions. Unfortunately, I can only discuss this issue, since my lab is closed due to my retirement.

We tried to consider the suggestion in the discussion:

518ff: Given the larger effect of GR in combination with NADPH on the redox state of GRXC2 as compared to the redox state of ROXY9 (Fig. 5), we assume that addition of these compounds to the assay (Bedhomme et al., 2012) would increase the difference between the de-glutathionylation activities of both proteins.

8) The enzymatic assay for GAPDH is extremely complicated, and the authors should consider that BPGA is an unstable compound. Why not use a BPGA-generating system (PGK plus ATP and 3PGA) in the cuvette also containing GAPDH and NADH? This system is normally used and, in my personal experience, is extremely efficient. Another tip concerns enzyme measurement. It is not optimal to proceed to obtain deltaAbs data only every minute. For this I recommend using a classical UV/Vis spectrophotometer and not a plate reader as I think was used.

Since we added the whole reaction mixture, including PGK and ATP and 3PGA (**line 724**), to the assay, the regenerating system is present in the GAPDH system. From our graphs, we did not have the impression that we needed a better resolution. In view of the many controls and replicates, a plate reader seemed to be more convenient considering that each measurement took one hour. We added the plate reader in the Methods section (**lines 726**).

9) For the purpose of further specifying the redox mechanism of GRX3 in *Chlamydomonas reinhardtii*, the authors should consider that during the process of de-glutathionylation of photosynthetic GAPDH (GAPA), an intramolecular disulfide bridge is formed after the transfer of glutathione moiety from the protein target to GRX3, which is subsequently reduced by the FDX/TRX system.

Thank you for this information. However, since we performed the GAPDH assay with rabbit GAPDH (now indicated in **line 710**), we think that we do not have to discuss this. In principle, we included the GAPDH de-glutathionylation assay in this manuscript in order to show that ROXY9 is not entirely inactive. I agree that this raises some unanswered questions on the mechanism, but the data set is reproducible. However, the activity is very weak and it is questionable whether it is relevant in vivo.

10) Line 715. Remove GSSG as it was not used

Since GSSG is formed when GSH and H₂O₂ are added (Abendinzadeh, et al. 1998), I consider it correct to leave it as it is.